# Asymptotically Optimal Quantile Pure Exploration for Infinite-Armed Bandits

**Xiao-Yue Gong**
Carnegie Mellon University
`exiaoyue@andrew.cmu.edu`

**Mark Sellke**
Harvard University
`msellke@fas.harvard.edu`

## Abstract

We study pure exploration with infinitely many bandit arms generated i.i.d. from an unknown distribution. Our goal is to efficiently select a single high quality arm whose average reward is, with probability $1 - \delta$, within $\varepsilon$ of being with the top $\eta$-fraction of arms. For fixed confidence, we give an algorithm with expected sample complexity $O\left(\frac{\log(1/\delta)\log(1/\eta)}{\eta\varepsilon^2}\right)$ which matches a known lower bound up to the $\log(1/\eta)$ factor. In particular the $\delta$-dependence is optimal and closes a quadratic gap. For fixed budget, we show the asymptotically optimal sample complexity as $\delta \to 0$ is $\log(1/\delta)\big(\log\log(1/\delta)\big)^2/c$. The value of $c$ depends explicitly on the problem parameters (including the unknown arm distribution) through a certain Fisher information distance. Even the strictly super-linear dependence on $\log(1/\delta)$ was not known and resolves a question of [GM20].

## 1 Introduction

In many learning problems, one faces the classical exploration versus exploitation tradeoff. A central example is the (stochastic) multi-armed bandit [LR85, BF85], where an agent is presented with a set of arms each of which when played gives a stochastic reward from an unknown and arm-dependent distribution. The performance of a bandit algorithm is most commonly determined by its *regret*, i.e. the difference between its average reward and the expected reward from the best arm. Multi-armed bandits and extensions have been applied in many settings including medical trials [BE95], online advertising [LCLS10], cognitive radio [AMTS11], and information retrieval [LPB17]. Optimal algorithms for the multi-armed bandit, including UCB, Thompson sampling, EXP3, and various forms of mirror descent, all make a principled tradeoff between exploration and exploitation.

In this work we focus on *pure exploration* bandit problems, a setting motivated by situations where the learning procedure consists of an *initial exploration* phase followed by a *choice* of policy to deploy. This is the case in hyperparameter optimization [GM20, LJD$^+$17] as well as reinforcement learning from simulated environments. As there is no longer a competing need to exploit, optimal algorithms for pure exploration differ from the more common regret setting.

Pure exploration problems were introduced in [EDMM02, MT04, EDMM06] in the probably-approximately-correct (PAC) model. Here given $K$ arms, one adaptively obtains samples until choosing one of the arms to output – the goal is to ensure that with probability $1 - \delta$, this arm has average reward within $\varepsilon$ of the best arm, with minimum possible sample complexity depending on $\varepsilon$ and $\delta$. The early works above focused on the *fixed confidence* setting in which one aims to minimize the expected sample complexity. Many subsequent works have also considered the *fixed budget* problem where the sample complexity is *uniformly* bounded.

While sharp results are known for pure exploration and other bandit problems with $K$ arms, for many applications such as advertising there are far too many arms to explore. This motivated the study of infinite-armed bandit problems in e.g. [BCZ$^+$97, WAM08]; the pure exploration version

was first studied in [AAKA18]. The main contribution of our work is to obtain near-optimal sample complexity for pure exploration problems with infinitely many arms in both the fixed confidence and fixed budget settings.

## 1.1 Problem Formulation

We now precisely formulate the infinite-armed pure exploration setting. Let $\mathcal{S} = \{a_1, a_2, \dots\}$ be a countably infinite set of stochastic bandit arms indexed by $i = 1, 2, \dots$. When an arm $i$ is sampled, it returns a $\{0, 1\}$-valued reward with mean $p_i$. The values $p_i$ are drawn i.i.d. from an arbitrary reservoir distribution $\mu$ supported in $[0, 1]$ (which is unknown to the player). We define the cumulative distribution function

$$G_\mu(\tau) = \mathbb{P}^{p \sim \mu}[p \leq \tau]$$

of $\mu$, and its (left-continuous) inverse

$$G_\mu^{-1}(p) = \inf\{\tau : G(\tau) \geq p\}.$$

Finally let $\mu^* = G_\mu^{-1}(1)$ denote the essential supremum of $\mu$, i.e. the maximum of its support.

An algorithm $\mathcal{A}$ interacts with $\mathcal{S}$ in the following way. At each time step $t = 1, 2, \dots, T$ the algorithm samples an arm $i_t \in \mathcal{S}$, and observes a corresponding Bernoulli reward $r_t \sim \text{Ber}(p_{i_t})$. The reward $r_t$ is independent of previous actions and feedback. Eventually at some time $T$, $\mathcal{A}$ chooses an arm $a_{i^*}$ to output. If the time-horizon $T = N$ is fixed, we say $\mathcal{A}$ has a *fixed budget* constraint. If $\mathbb{E}[T] \leq N$ is bounded only in expectation, we say $\mathcal{A}$ has a *fixed confidence* constraint.

For $\eta, \varepsilon, \delta > 0$, we say $\mathcal{A}$ is $(\eta, \varepsilon, \delta)$-PAC if

$$\mathbb{P}\big[p_{i^*} \geq G^{-1}(1 - \eta) - \varepsilon\big] \geq 1 - \delta \tag{1.1}$$

and set

$$\alpha \equiv G^{-1}(1 - \eta)$$

to be the target quantile value. We emphasize that while $\eta$ is known, $\alpha$ may not be as it depends on the unknown $\mu$. The definition (1.1) stems from [AAKA18]. As brief justification for the parameter $\eta$, note that in an infinite-armed setting the reservoir $\mu$ could give $\varepsilon$-optimal arms with arbitrarily small probability. Thus it is impossible to give a non-asymptotic classical $(\varepsilon, \delta)$-PAC in our setting without assumptions on $\mu$. Taking $\eta > 0$ as above ensures that a positive fraction of arms are "good enough" and will enable such guarantees. One also cannot set $\varepsilon$ to zero: for example if $\mu$ is supported in $[0.5 - e^{-N^4}, 0.5]$, the quantile value of the output arm will be essentially uniform for any $N$-sample algorithm.

The purpose of this paper is to give $(\eta, \varepsilon, \delta)$-PAC algorithms whose sample complexity $N$ is minimal. We can now state our main results. Let us emphasize that unless explicitly stated, no assumptions on the reservoir distribution $\mu$ are made, nor does the algorithm have any prior knowledge about $\mu$. Our main result in the fixed confidence case is as follows.

**Theorem 1.1.** *For any $(\eta, \varepsilon, \delta)$, there exists a $(\eta, \varepsilon, \delta)$-PAC algorithm with expected sample complexity $O\left(\frac{\log(1/\delta)\log(1/\eta)}{\eta\varepsilon^2}\right)$.*

In the fixed budget setting, our interest is especially in the high-confidence regime $\delta \to 0$, where we obtain the following. The following statement is a slightly informal combination of Theorems 3.1 and 3.2. We note that while the main statement requires $\alpha$ to be given, this is not essential in several cases as discussed extensively in the Appendix. For example if $\alpha \geq \frac{1+\varepsilon}{2}$ then $\alpha$ does not need to be given.

**Theorem 1.2** (Informal). *For any fixed $(\eta, \varepsilon)$, let $\alpha = G^{-1}(1 - \eta)$ be given and set $\beta = \alpha - \varepsilon$. Then as $\delta \to 0$, the optimal $(\eta, \varepsilon, \delta)$-PAC algorithm under fixed budget has sample complexity*

$$N = \left(c_{\alpha,\beta}^{-1} \pm o_{\delta \to 0}(1)\right)\log(1/\delta)\big(\log\log(1/\delta)\big)^2;$$

$$c_{\alpha,\beta} \equiv \frac{\big(\arccos(1 - 2\alpha) - \arccos(1 - 2\beta)\big)^2}{2} \tag{1.2}$$

$$= \frac{\left(\int_\beta^\alpha \frac{dx}{\sqrt{x(1-x)}}\right)^2}{2}. \tag{1.3}$$

An equivalent statement is that given exactly $N$ samples, the optimal failure probability $\delta$ to have $p_{i^*} \geq G^{-1}(1 - \eta) - \varepsilon$ decays as $\exp\left(-\frac{N(c \pm o(1))}{\log^2(N)}\right)$. Indeed once $\eta$ and $\varepsilon$ are fixed, the question of minimizing the sample complexity $N = N(\delta)$ (given a target confidence $\delta$) is equivalent to minimizing the failure probability $\delta = \delta(N)$ (given a sample complexity $N$). These viewpoints are equivalent in both settings we study, and we switch between them at times.

Interestingly the value $\eta$ makes no appearance in Theorem 1.2, so it is asymptotically irrelevant for the $\delta \to 0$ regime of fixed budget pure exploration. In fact the value $\arccos(1 - 2\alpha) - \arccos(1 - 2\beta)$ appearing in the definition of $c_{\alpha,\beta}$ is the Fisher-information distance between $\alpha$ and $\beta$ in the exponential family of Bernoulli random variables via the formula (1.3). See just below Theorem 3.1 for a brief explanation of why $\eta$ does not enter the asymptotic sample complexity.

**Remark 1.1.** *In our problem formulation above we assumed rewards are Bernoulli, i.e. lie in $\{0, 1\}$. In fact as long as the quality of arm $i$ is measured by its mean reward, this assumption loses no generality and is just a technical convenience: our results extend verbatim to $[0, 1]$-valued rewards.*

*Indeed any arm with $[0, 1]$-valued rewards can be transformed into a Bernoulli arm with $\{0, 1\}$-valued rewards and the same mean: simply turn reward $r \in [0, 1]$ into reward 1 with probability $r$, and 0 with probability $1 - r$. Note that this reduction (used also in Section 1.2 of [AG12]) might increase the instance-dependent sample complexity of some reservoir distributions, but our results only refer to the distribution of arm means under the reservoir which is unchanged by the reduction.*

## 1.2 Further Notation

We use the convention that algorithms collect 1 sample per unit time until terminating, so the time $t$ equivalently denotes the number of total samples collected so far. Denote by $n_{i,t}$ the number of samples of arm $a_i$ collected by time $t$. The $n$-th time $a_i$ is sampled, its reward is $r_{i,n} \in [0, 1]$. The total reward of arm $i$ up to time $t$ is

$$R_{i,t} = \sum_{n=1}^{n_{i,t}} r_{i,n}.$$

The corresponding average reward is $\hat{p}_{i,t} = \hat{p}_i(n_{i,t}) = \frac{R_{i,t}}{n_{i,t}}$. We use $C$ to indicate a universal constant independent of all parameters in this paper, and $o_n(1)$ and $o_N(1)$ to denote quantities tending to 0 as $n \to \infty$ or $N \to \infty$, with other parameters implicitly held constant. However in Section C we use e.g. $\Omega_{\alpha,\varrho}$ to indicate an asymptotic lower bound with implicit constant factor depending on the values of $\alpha, \varrho$, which are treated as fixed. In all our uses of these notations it is $n$ or $N$ which is tending to infinity while other parameters are always treated as fixed.

## 1.3 Related Work

As discussed above, this work belongs to the area of *pure exploration* for multi-armed bandit problems. Unlike ordinary bandit problems where one aims to minimize the regret compared to the best arm [BCB12, Sli19], in pure exploration all that matters is the final arm selected by the algorithm. We survey several existing results below, with an emphasis on the high-probability regime of small $\delta$. See e.g. Chapter 33 of [LS20] for a more detailed survey.

Pure exploration was first studied in [EDMM02, MT04, EDMM06] in the probably-approximately-correct model. Here given $K$ arms, one adaptively obtains samples until choosing one of the arms to output – the goal is to ensure that with probability $1 - \delta$, this arm has average reward within $\varepsilon$ of the best arm. These works showed that the optimal fixed confidence sample complexity is $\Theta\left(\frac{K}{\varepsilon^2} \log \frac{1}{\delta}\right)$.

Later, [BMS09] considered the *simple regret* of pure exploration problems, namely the regret incurred at the final timestep. [ABM10] studied the closely related problem of identifying the best arm, obtaining nearly tight sample complexity bounds in terms of the the sum of the squared inverse suboptimality-gaps $H = \sum_{i \neq i^*} \Delta_i^{-2}$. Further upper and lower sample complexity bounds have been obtained in several works. For example [CL15, KCG16] show that for fixed confidence, the sample complexity scales as $\Theta\left(H \log(1/\delta)\right)$ as $\delta \to 0$. The fixed budget setting, in which the number of adaptive samples is upper-bounded *almost surely* rather than in expectation, turns out to be more difficult. [CL16] proved that the optimal fixed budget sample complexity can be $\Theta\left(H \log(K) \log(1/\delta)\right)$ as $\delta \to 0$, i.e. the fixed budget constraint may lead to an additional $\log(K)$

factor. However it reverts to $\Theta\left(H\log(1/\delta)\right)$ when the value of $H$ is known beforehand. Many recent works have studied other aspects of pure exploration, for example by incorporating structured feedback; see [JMNB14, CGL+17, KSJ20, KG21, TRMD21, ACD21, ZKSN22, AAJ+22].

Infinite-armed bandits have also much received previous study, e.g. [BCZ+97, WAM08]. Since near-optimal arms may be arbitrarily rare, it is natural to instead compare with a **quantile** of the arm distribution. For example [CK18] aims to minimize regret relative to such a quantile.

The $(\eta, \varepsilon, \delta)$-PAC guarantees we address in this paper were first studied in [AAKA18], for infinitely many arms in the fixed confidence setting. Their approach was to sample $K \asymp \frac{\log(1/\delta)}{\eta}$ arms and then apply a PAC algorithm for $K$-armed pure exploration. As discussed at the beginning of Section 2, the resulting algorithm "pays twice" for the high confidence level $1 - \delta$ which leads to a suboptimal $O(\log^2(1/\delta))$ sample complexity upper bound. Top-$k$ extensions were also studied in [RLS19, CK19]; the $\log^2(1/\delta)$ scaling is still present in their results.

Of particular note is the work [dHCMC21] which considers also both fixed budget and confidence settings and obtains somewhat similar looking results. However they restrict attention to a special class of reservoir distributions with supremum achieved by an atom, which must be $\Delta$-larger than the rest of the support. This structural assumption of a $\Delta$-gap intrinsically reduces fixed budget sample complexity: their result (see Theorem 4 therein) is actually better than the lower bound we show in Theorem 3.2 as there is no appearance of $\log\log(1/\delta)$ (i.e. $\log(T)$ in their notation).

From their fixed budget estimate, [dHCMC21] deduce (at the end of Section 1 therein) the same bound as Theorem 1.1 in their setting for the special case $\varepsilon = \Delta$. Our Theorem 3.2 shows that for the general reservoirs we consider, passing from fixed budget to fixed confidence is inherently suboptimal: the factors of $\log\log(1/\delta)$ would remain, but are extraneous for fixed confidence. This underscores that Theorem 1.1 is genuinely new despite the superficial similarity with [dHCMC21], since their proof cannot work in our setting.

Finally [GM20] studied the infinite-arm pure exploration problem where $\alpha$ is given, also focusing on the $\delta \to 0$ asymptotics. They proposed an algorithm with fixed budget sample complexity $\mathcal{O}\left(\log(1/\delta)\left(\log\log(1/\delta)\right)^2\right)$, and asked whether the $\log\log(1/\delta)$ factors are necessary. Theorem 3.2 shows their bound is optimal up to constant factors in terms of $\delta$ and in fact obtains the tight constant. Interestingly [GM20] were motivated by complexity theoretic applications to *amplification* and *derandomization*, where bandit arms correspond to random seeds.

We remark that the analysis in [GM20] seems to be technically incomplete. In particular in Lemma 4.5 of (the cited, journal version of) the paper, they neglect to take a union bound over sequences $(T_1, \ldots, T_k)$ summing to $T$ but only estimate the probability of each fixed sequence $(T_1, \ldots, T_k)$. This is a serious gap since the number of such sequences is exponentially large in $T$. However their idea to use a moving sequence of rejection thresholds was fundamentally correct. It is similar to the main phase of our Algorithm 3, for which we give a fully rigorous, supermartingale-based analysis.

## 2 The Fixed Confidence Setting

Our fixed confidence algorithms proceeds in two phases. The first phase aims to estimate the target quantile value, which we recall depends on the unknown $\alpha$. The second phase then aims to find a single arm which is almost as good as this estimate with high probability.

Focusing on the $\delta$-dependence, a challenge with infinitely many arms is that to succeed with probability $1 - \delta$, it is necessary both to sample $\log(1/\delta)$ arms to ensure a good arm is ever observed, and to sample an arm $\log(1/\delta)$ times before outputting it as $i^*$. This is why the approach of [AAKA18] requires $O(\log^2(1/\delta))$ samples: they obtain $O(\log(1/\delta))$ samples each of $O(\log(1/\delta))$ arms. However in our algorithm, the first phase samples $O(\log(1/\delta))$ arms $O(1)$ times each, while the second phase samples $O(1)$ arms $O(\log(1/\delta))$ times each. This allows us to satisfy both necessary conditions above without paying twice for the confidence level.

### 2.1 The Algorithm for Fixed Confidence

We first give in Alg. 1 a simple procedure to estimate the top $\eta$ quantile, allowing an $\varepsilon/3$ error as well as an $\eta/2$ error in the quantile itself. Alg. 1 obtains $O\left(\frac{\log(1/\eta)}{\varepsilon^2}\right)$ samples from each of the

---
**Algorithm 1:** Output $\hat{\alpha} \in \left[ G^{-1}(1-\eta) - \frac{\varepsilon}{3}, G^{-1}\left(1 - \frac{\eta}{2}\right) + \frac{\varepsilon}{3} \right]$ with probability $1 - \frac{\delta}{2}$
---
**1** **input:** arm set $\mathcal{S} = (a_1, a_2, \dots)$ and parameters $\eta, \varepsilon, \delta \in (0, 1)$.
**2** initialize: $K = \frac{C \log(1/\delta)}{\eta}$.
**3** **for** $i = 1, 2, \dots, K$ **do**
**4** $\quad$ Collect $n = \frac{C \log(1/\eta)}{\varepsilon^2}$ samples of arm $i$. Set $\hat{p}_i = \hat{p}_i(n)$ to be the average observed reward.
**5** **end**
**6** Let $\hat{\alpha}$ be the $k$-th largest value in $\{\hat{p}_1, \dots, \hat{p}_K\}$ for $k = \left\lceil \frac{3K\eta}{4} \right\rceil$.
**7** Return $\hat{\alpha}$
---

---
**Algorithm 2:** Output $a_{i^*}$ such that $p_{i^*} \geq \hat{\alpha} - \varepsilon$ with probability $1 - \delta$.
---
**1** **input:** arm set $\mathcal{S} = (a_1, a_2, \dots)$ and parameters $(\eta, \varepsilon, \delta, \hat{\alpha})$
**2** **for** $i = K+1, K+2, \dots, K + \frac{C \log(1/\delta)}{\eta}$ **do**
**3** $\quad$ Collect $\frac{C \log(1/\eta\delta)}{\varepsilon^2}$ samples of arm $i$. Set $\hat{p}_i$ to be the average reward.
**4** $\quad$ **if** $\hat{p}_i \geq \hat{\alpha} - \frac{\varepsilon}{3}$ **then**
**5** $\quad\quad$ Return $a_i$
**6** $\quad$ **end**
**7** **end**
---

first $K = O\left( \frac{\log(1/\delta)}{\eta} \right)$ arms $a_1, \dots, a_K$. The resulting estimator $\hat{\alpha}$ is the $1 - \frac{3\eta}{4}$ quantile of the empirical average rewards $\hat{p}_1, \dots, \hat{p}_k$. Its main guarantee is below.

**Proposition 2.1.** *Fix $0 \leq \eta, \varepsilon, \delta \leq 1$. With probability at least $1 - \frac{\delta}{2}$, the output $\hat{\alpha}$ of Alg. 1 satisfies*

$$\hat{\alpha} \in \left[ G^{-1}(1-\eta) - \frac{\varepsilon}{3}, G^{-1}\left(1 - \frac{\eta}{2}\right) + \frac{\varepsilon}{3} \right].$$

*Moreover, Alg. 1 has sample complexity*

$$O\left( \frac{\log(1/\eta) \log(1/\delta)}{\eta \varepsilon^2} \right).$$

Next Alg. 2 repeatedly chooses a new arm $a_i$ and obtains $O\left( \frac{\log(1/\eta\delta)}{\varepsilon^2} \right)$ samples. It accepts if the sample mean was at least $\hat{\alpha} - \frac{\varepsilon}{3}$, and otherwise moves on to the next arm. If $\frac{C \log(1/\delta)}{\eta}$ arms have been tried without success, then Alg. 2 outputs no arm, thus declaring failure. This termination condition is necessary to avoid incurring huge sample complexity when Alg. 1's estimate $\hat{\alpha}$ of $\alpha$ is inaccurate. Since typically $\Omega(\eta)$ of arms will be good enough to usually succeed, this also preserves the $1 - \delta$ confidence level. We now give the following more detailed restatement of Theorem 1.1.

**Theorem 2.1.** *Apply Algorithm 1 followed by Algorithm 2 using the resulting value $\hat{\alpha}$. This combined algorithm has expected sample complexity $O\left( \frac{\log(1/\eta) \log(1/\delta)}{\eta \varepsilon^2} \right)$. Moreover its output $a_{i^*}$ satisfies*

$$\mathbb{P}[p_{i^*} \geq G^{-1}(1-\eta) - \varepsilon] \geq 1 - \delta.$$

Proposition 2.1 and Theorem 2.1 are proved in Appendix A. We note that in analyzing Algorithm 2, typically the returned arm $a_i$ has $i \leq K + O(1/\eta)$. This is because each arm $a_i$ in the top $\eta/2$ quantile has a good chance to pass the test of Algorithm 2. In particular, the expected sample complexity calculation never multiplies two $\log(1/\delta)$ terms together. However, continuing for $K + \frac{C \log(1/\delta)}{\eta}$ steps is important to ensure a $1 - \delta$ success probability. Algorithm 2 stops after $O(\log(1/\delta)/\eta)$ arms instead of continuing forever in order to guard against erroneous estimates from Algorithm 1.

**Remark 2.1.** *Our fixed confidence algorithm, given by combining Alg. 1 with Alg. 2 as above, requires only $O(1)$ **batches** in expectation. Here a batched algorithm operates in a small number of batched phases. At the start of each phase, such an algorithm chooses $b$ arms to sample exactly $s$ times each, where $b, s$ can both depend adaptively on the previous feedback, but cannot be*

*changed during the current phase. Minimizing the number of required batches is often desirable, see e.g. [PRCS16, GHRZ19]. In particular Alg. 1 uses a single batch with $s_1 = \frac{C \log(1/\eta)}{\varepsilon^2}$ samples of $b_1 = \frac{C \log(1/\delta)}{\eta}$ arms. Then Alg. 2 can be implemented in a batched way with $s_2 = \frac{C \log(1/\eta\delta)}{\varepsilon^2}$ and a sequence of batch sizes $b_{2,i} = \frac{2^i}{\eta}$ for $1 \leq i \leq \log_2(C \log(1/\delta))$. (In the latter phase, one stops after finding an arm to accept.) While this construction uses $O(1)$ batches **in expectation**, it could be interesting to explore pure exploration with an exactly fixed number of batches, which is more analogous to the fixed budget setting.*

## 2.2 Near-Optimality in the Fixed Confidence Regime

Here we explain why the guarantee of Theorem 2.1 is nearly optimal. For comparison, recall from the important work [MT04] that $\Theta\left(\frac{K \log(1/\delta)}{\varepsilon^2}\right)$ samples are necessary and sufficient for $(\varepsilon, \delta)$-PAC pure exploration in the $K$-armed bandit problem with fixed confidence. Intuitively, one expects this problem to be related to ours via $\eta \approx 1/K$. In fact the following infinite-arm analog was later shown.[1] It follows that the guarantee of Theorem 1.1 is optimal up to the $\log(1/\eta)$ factor.

**Proposition 2.2** (Theorem 1 and Remark 2 of [AAKA18])**.** *There exists an absolute constant $c > 0$ such that the following holds. For any $1/4 \leq \alpha \leq 3/4$ and $\eta, \delta \leq 1/10$ and for any pure exploration algorithm $\mathcal{A}$ with expected sample complexity $N \leq \frac{c \log(1/\delta)}{\eta\varepsilon^2}$, there exists a reservoir distribution such that $\mathcal{A}$ fails to be $(\eta, \varepsilon, \delta)$-PAC.*

Letting $L = \frac{\log(1/\delta)}{\eta\varepsilon^2} \geq 1/\eta$ be the lower bound from Proposition 2.2, the expected sample complexity of our algorithm is at most $O(L \log(1/\eta)) \leq O(L \log L)$. Hence Theorem 1.1 is always nearly optimal compared to the lower bound. Prior to our work there was a quadratic gap as the best upper bound for general reservoirs (Theorem 6 in [AAKA18]) was proportional to $\log^2(1/\delta)$. We note also that Theorem 5 of [dHCMC21] showed a similar result to Proposition 2.2.

## 3 Fixed Budget

Our algorithm and lower bounds for the fixed budget setting are much more technical, with full details given in the Appendix. Here we carefully state the results and give outlines of the proofs.

## 3.1 Precise Results for Fixed Budget

We first state the results in the easier case that $\alpha$ is given. Theorems 3.1 and 3.2 below give rigorous statements of Theorem 1.2. As there, we will write $\beta$ for the target value $\alpha - \varepsilon$ when $\alpha$ is given, since then the value $\varepsilon$ plays no role.

**Theorem 3.1.** *For any fixed $0 < \beta < \alpha < 1$, there is a sequence $(\mathcal{A}_N)_{N \geq 1}$ of $N$-sample algorithms given explicitly by Algorithm 3 such that for any $\eta \in (0, 1)$ and any sequence of reservoir distributions $\mu_N$ with $G_{\mu_N}^{-1}(1 - \eta) \geq \alpha$, with $c_{\alpha,\beta}$ as in (1.2):*

$$\limsup_{N \to \infty} \frac{(-\log \mathbb{P}_{\mu_N}[p_{i^*} < \beta]) \cdot \log^2 N}{N} \geq c_{\alpha,\beta}. \tag{3.1}$$

Conversely, the following lower bound applies for any quantile $\eta \in (0, 1)$, and holds even when $\alpha$ is known (which only makes pure exploration easier). It implies that $\eta$ is asymptotically irrelevant for fixed budget sample complexity, i.e. the sample complexity of approximating the $\eta = 0.01$-quantile and $\eta = 0.99$ quantile in fixed budget pure exploration depends only on the quantile values themselves as $\delta \to 0$. Some intuition for this is as follows. To succeed in pure exploration, one should have sampled the eventually outputted arm at least $\Omega(\log 1/\delta)$ times. The main obstacle to success in the fixed budget case is that any arm we obtain many samples of might gradually degrade over time. The probability of this degradation is essentially given by small probabilities coming from Chernoff-bound type events, which dominate the prior probability that the arm is in a top quantile.

---

[1][AAKA18] states the result more generally. Specializing to $\mathbb{P}[p_i = \alpha] = \eta$ and $\mathbb{P}[p_i = \alpha - \varepsilon] = 1 - \eta$ for any $1/4 \leq \alpha \leq 3/4$ recovers the statement of Proposition 2.2.

**Theorem 3.2.** *For any $0 < \eta < 1$ and $0 < \beta < \alpha < 1$ there exists a sequence of reservoir distributions $\mu_N$ with $\alpha = G_{\mu_N}^{-1}(1 - \eta)$ such that for any sequence of $N$-sample algorithms $\mathcal{A}_N$,*

$$\liminf_{N \to \infty} \frac{(-\log(\mathbb{P}_{\mu_N}[p_{i^*} < \beta])) \cdot \log^2 N}{N} \leq c_{\alpha,\beta}. \tag{3.2}$$

**Remark 3.1.** *As discussed in the Appendix, the requirement that $\alpha$ be known in Theorem 3.1 is technically necessary to deal with potential discontinuity of $\alpha$ as $\eta$ varies, but can be removed under mild conditions. In Theorems B.1, B.2, and B.3 below we give three concrete formulations under which the guarantee (3.1) can be achieved without knowledge of $\alpha$. Informal descriptions (any 1 of which suffices on its own) are:*

1. *$\mu_N$ obeys $G_{\mu_N}^{-1}(1 - \eta) \geq \frac{1+\varepsilon}{2}$.*

2. *$\alpha$ is redefined as the average of $G_\mu^{-1}(\eta)$ for $\eta$ ranging over an interval.*

3. *$\mu_N = \mu$ is independent of $N$, and the target value is*
$$\beta = \mu^* - \varepsilon_1$$
   *for fixed $\varepsilon_1 > \varepsilon$. (Recall $\mu^*$ is the maximum value in the support of $\mu$.)*

**Remark 3.2.** *In fact (see Theorem C.9 in the Appendix), Theorem 3.1 holds even if the algorithms $\mathcal{A}_N$ must output $\log N$ distinct arms $i_1^*, \ldots, i_{\log N}^*$, all of which must satisfy $p_{i_j^*} \geq \beta$. I.e. for suitable $(\mathcal{A}_N)_{N \geq 1}$,*

$$\limsup_{N \to \infty} \frac{(-\log \mathbb{P}\left[\min_{j \in [\log N]} p_{i_j^*} < \beta\right]) \cdot \log^2 N}{N} \geq c_{\alpha,\beta}.$$

### 3.2 Algorithm for Fixed Budget

Here we present our Algorithm 3 for the fixed budget problem. We note that this algorithm is given as input the value of $\alpha$. See Appendix C.1 for first-stage algorithms which estimate $\alpha$ with sufficiently high accuracy and low sample complexity in the three scenarios of Remark 3.1. We use an explore-and-discard approach: at each time, it is currently exploring some arm $i$, having permanently discarded arms $1, 2, \ldots, i - 1$ and not yet interacted with arms $i + 1, i + 2, \ldots$. It turns out to be very convenient to operate in a "B-batch-compressed" way for an increasing integer sequence $B = (b_1, b_2, \ldots)$. This means that at all times, the current arm $i$ has been sampled $b_k$ times for some $k$, and the samples between $b_k$ and $b_{k+1}$ are compressed into a single decision. The sequence $b_k$ increases geometrically at a small rate, i.e. $b_{k+1}/b_k \approx 1 + \varrho$ for small $\varrho$ once $k$ is mildly large. This batch-compression turns out to be without loss of generality up to a factor $1 + \varrho$ in the sample complexity, and helps the analysis operate on the proper geometric time-scales.

We now give a precise description as well as pseudo-code. Let $0 < \varrho \ll \varrho_1 \ll \varrho_2 \ll 1$ be suitably small constants (i.e. choose $\varrho_2$ sufficiently small, then $\varrho_1$, then $\varrho$). We define $b_k$ and other parameters as follows:

$$
\begin{aligned}
\theta(a) &= \arccos(1 - 2a) \quad \forall\, a \in [0, 1], \\
b_0 &= \lceil \varrho_1 \log^2(N) \rceil, \\
k_0 &= \lceil \log_{1+\varrho}\left(\log^4(N)/b_0\right) \rceil \\
b_k &= \lceil b_0(1 + \varrho)^k \rceil, \quad k \leq k_0 \\
b_{k_0+j} &= \lceil (1 + \varrho)^j b_{k_0} \rceil, \quad j \geq 1 \\
\tau_k &= \alpha - \varrho - \frac{k}{\sqrt{\log N}}, \quad k \leq k_0 \\
\tau_{k_0+j} &= \theta(\alpha - 2\varrho) - j \cdot \frac{(\theta(\alpha) - \theta(\beta))\varrho(1 - \varrho_2)}{\log N}, \quad j \geq 1.
\end{aligned}
\tag{3.3}
$$

The outer loop dictates the arm $i$ under consideration. While exploring arm $i$, the main phase (shown in the last for loop) consists of collecting a new batch of $b_{k_0+j} - b_{k_0+j-1} \approx \varrho b_{k_0+j-1}$ samples, and rejecting if the new empirical mean reward $\hat{p}_{i,b_{k_0+j}}$ drops below

$$\theta^{-1}\left(\theta(\alpha - 2\varrho) - j \cdot \frac{(\theta(\alpha) - \theta(\beta))\varrho(1 - \varrho_2)}{\log N}\right).$$

---

**Algorithm 3:** Output arm with $p_i \geq \beta$ using $N$ samples with high probability

---

**1** **input:** parameters $N, \alpha, \beta$, and an infinite sequence of arms $i = 1, 2, \ldots$
**2** initialize: parameters from (3.3) and $i = 0$
**3** **while** *fewer than $N$ samples have been collected* **do**
**4** $\quad$ $i \leftarrow i + 1$
**5** $\quad$ Get $b_0$ samples of arm $i$.
**6** $\quad$ **if** $\hat{p}_{i,b_0} \leq \alpha - \varrho$ **then**
**7** $\quad\quad$ **Reject** arm $i$
**8** $\quad$ **end**
**9** $\quad$ **for** $k = 1, 2, \ldots, k_0$ **do**
**10** $\quad\quad$ Get $b_k - b_{k-1}$ samples of arm $i$ (total $b_k$).
**11** $\quad\quad$ **if** $\hat{p}_{i,b_k} \leq \alpha - \varrho - \frac{k}{\sqrt{\log N}}$ **then**
**12** $\quad\quad\quad$ **Reject** arm $i$;
**13** $\quad\quad$ **end**
**14** $\quad$ **end**
**15** $\quad$ **for** $j = 1, 2, \ldots$ **do**
**16** $\quad\quad$ Get $b_{k_0+j} - b_{k_0+j-1}$ samples of arm $i$ (total $b_{k_0+j}$).
**17** $\quad\quad$ **if** $\theta(\hat{p}_{i,b_{k_0+j}}) \leq \theta(\alpha - 2\varrho) - j \cdot \frac{\left(\theta(\alpha) - \theta(\beta)\right)\varrho(1-\varrho_2)}{\log N}$ **then**
**18** $\quad\quad\quad$ **Reject** arm $i$
**19** $\quad\quad$ **end**
**20** $\quad$ **end**
**21** **end**
**22** Return arm $i$.

---

Let us explain the point of this formula. First ignoring the function $\theta$ for now, we see that the rejection threshold for $\hat{p}_{i,b_k}$ steadily decreases with $k$. This threshold is tuned so that when arm $i$ has been sampled say $\Omega(N/\log N)$ times, we have $k \asymp \log(N)/\varrho$ which results in a threshold slightly larger than $\beta$. Thus an arm which survives for such a long time will be prepared for acceptance as a new-optimal arm.

This strategy can be motivated as follows. The chief worry in fixed budget exploration is that arms might slowly degrade after many samples have been invested into them, which suggests a gradually decreasing rejection threshold. We designed this threshold to drop by a constant divided by $\log(N)$ each time the number $b_k$ of samples doubles. This ensures that for arm $i$ to be rejected at time $b_{k+1}$, the last $b_{k+1} - b_k$ samples must have behaved atypically (compared to their past behavior) by roughly $\frac{\sqrt{b_{k+1}}}{\log(N)}$ standard deviations. The probability of this rare event is roughly $\exp(-Cb_{k+1}/\log^2(N))$ for some constant $C$ by a Chernoff bound. Therefore if all $N$ samples are used on eventually rejected arms, the product of these rare event probabilities will be roughly $\exp(-CN/\log^2(N))$ since each $b_{k+1}$ counts the number of samples used on an individual arm.

The use of the non-linear function $\theta$ above is essential to achieve the optimal constant $c_{\alpha,\beta}$ in Theorem 3.1. At a high-level, $\theta$ balances the $p(1-p)$-dependence of optimal Chernoff bounds on the underlying probability $p$ of the Binomial random variable. If the optimal constant is not desired, then Theorem 3.1 can be simplified somewhat; the nonlinear $\theta$ is not needed and one can double the number of samples at each step rather than multiplying by $1 + \varrho$ for small $\varrho$. The earlier for loops in Algorithm 3 are technically important to handle small sample sizes for each given arm, before the required Chernoff bounds have kicked in asymptotically.

### 3.3 Analysis of Algorithm 3

The analysis of Algorithm 3 goes by controlling the tail distribution for the rejection time of a given arm (this time is taken to be zero if no rejection ever happens). Intuitively, we are most worried about arms which slowly degrade, thus wasting many samples. The following lemma, proved in the Appendix by a supermartingale argument, is key to rigorize this idea.

**Lemma 3.** *Suppose $(Y_i)_{i \geq 1}$ are i.i.d. random variables with non-negative integer values, and $\mathbb{E}[Y_i^c] \leq 1$ holds for some constant $c \geq 0$. Then with*

$$M = \sup_{j \geq 0} \prod_{1 \leq i \leq j} Y_i$$

*we have $\mathbb{P}[M \geq A] \leq A^{-c}$.*

We apply Lemma 3 in the following way. Let $X_i$ be the number of samples used by arm $a_i$ before rejection, and $I_i \in \{0, 1\}$ be the indicator of the event that $a_i$ is ever rejected (even if Algorithm 3 were to continue past time $N$ and sample arm $i$ an infinite number of times). We set $Y_i = e^{X_i} \cdot I_i$. The bulk of the analysis thus reduces to proving that $\mathbb{E}[Y_i^c] \leq 1$ for a suitable exponent $c$. Once this is shown, Lemma 3 ensures that $M$ is small with high probability. Since $\log(M)$ is exactly the total number of samples used on eventually-rejected arms, if say $\log M \leq N \left(1 - \frac{1}{\log N}\right)$ then the last arm $i^*$ to be selected must have passed enough rejection thresholds to have $p_{i^*} \geq \beta$ with sufficiently high probability.

The early behavior of Algorithm 3 and $b_k$ are specifically designed so that small values of $Y_i$ contribute little in expectation. The tail behavior of $Y_i$ (corresponding to rejecting arm $i$ after a long time) is controlled by a technical analysis involving many adjacent time-scales in Subsection C.4 of the Appendix. Interestingly this tail analysis of $Y_i$ never explicitly models the reward probability $p_i$. This is because we are able to argue that a rejection requires the early and late time behaviors of arm $i$ to differ *from each other*, which is unlikely since the rewards form an i.i.d. sequence.

## 3.4 On the Lower Bound Proof

We consider the lower bound Theorem 3.2 to be the technical highlight of this paper. In it we face the challenge of proving a very *small* lower bound on the failure probability of any fixed budget algorithm. To do so we construct an **online adversary with bounded probabilistic strength** to distort rewards. The setup follows; recall $B$-batch-compressed algorithms as defined in Subsection 3.2.

**Definition 3.1.** *An **adaptive randomness distorting adversary** $\mathbb{A}$ interacts with a $B$-batch-compressed algorithm $\mathcal{A}$ in the following way. Suppose $\mathcal{A}$ chooses to increase the number of samples of arm $a_i$ from $b_k$ to $b_{k+1}$. Then $\mathbb{A}$ may restrict the set of possible outcomes of these $b_{k+1} - b_k$ samples. Additionally, when $\mathcal{A}$ outputs an arm $a_{i^*}$, the adversary can restrict the possible values of $p_{i^*}$.*

We will refer to adversarial actions as *declarations*. Thus when $\mathcal{A}$ chooses a batch of samples, $\mathbb{A}$ may declare that some property holds for the observed rewards.

As defined above, an adversary $\mathbb{A}$ can do anything. We will limit the power of $\mathbb{A}$ to make **low-probability** declarations. To formalize this, we charge $\mathbb{A}$ per "bit" of probabilistic distortion, and give $\mathbb{A}$ a deterministic "budget" for doing so. We measure this budget according to the algorithm's filtration which we refer to the "jointly Bayesian" viewpoint. One should think that the reservoir distribution $\mu_N$ is known to both $\mathcal{A}$ and $\mathbb{A}$, but neither has any information on the true reward probabilities $p_i$ beyond the observed rewards. Thus $\mathcal{A}$ and $\mathbb{A}$ share at any time $t$ the posterior distribution $\boldsymbol{\mu}^t$. In particular recalling (B.9), $\boldsymbol{\mu}^t$ determines the distribution for the outcome of the next batch of $b_{k+1} - b_k$ samples.

**Definition 3.2.** *Suppose that at time $t$, the declaration of $\mathbb{A}$ has probability $P_t$ to hold according to $\boldsymbol{\mu}^t$. Let the sum*

$$\mathsf{Cost}_t = \sum_{s \leq t} \log(1/P_s) \tag{3.4}$$

*be the total cost of $\mathbb{A}$ up to time $t$, and $\mathsf{Cost}_N$ the total cost of $\mathbb{A}$. We say $\mathrm{strength}(\mathbb{A}) \leq \mathsf{Cost}$ holds for some $\mathsf{Cost} \in \mathbb{R}$ if the bound $\mathsf{Cost}_N \leq \mathsf{Cost}$ almost surely.*

The next key lemma shows that to obtain a lower bound for the failure probability of an algorithm, it suffices to prevent success using a low strength $\mathbb{A}$.

**Lemma 4.** *Suppose there is a randomness distorting adversary $\mathbb{A}$ of strength $\mathsf{Cost}$ whose declarations ensure that any algorithm $\mathcal{A}$ outputs $i^*$ satisfying $p_{i^*} \leq \beta$ almost surely. Then the true failure probability of $\mathcal{A}$ is*

$$\mathbb{P}^{\mu_N, \mathcal{A}}[p_{i^*} \leq \beta] \geq e^{-\mathsf{Cost}}.$$

*Proof.* At each step $t$, let the random variable $E_t$ denote the minimum possible conditional probability of the event $p_{i^*} \leq \beta$ for *any* algorithm (in the algorithm's jointly Bayesian filtration). We claim that conditioned on $\mathbb{A}$'s declarations holding, for any $\mathcal{A}$ the quantity $M_t \equiv E_t \prod_{s \leq t} P_s$ evolves as a supermartingale in this filtration. This suffices because it implies $E_0 = M_0 \geq \mathbb{E}[M_T] \geq e^{-\text{Cost}}$. (Here $T$ is the random number of total batches used.)

Indeed suppose we are at time $t$ and the next batch has been declared but not sampled. (Identical arguments apply when the adversary restricts $p_{i^*}$ in the last stage.) Let the $\sigma$-field $\mathcal{F}_t$ denote all information up to this point including the declaration of the next batch. Let $\mathbb{E}$ denote an expectation where the samples from the next batch is distributed according to $\boldsymbol{\mu}^t$. Let $\widetilde{\mathbb{E}}$ denote an expectation where $\mathbb{A}$'s declaration is conditioned to hold on the next batch. The dynamic programming principle implies that

$$\mathbb{E}[E_{t+1} \mid \mathcal{F}_t] \leq E_t$$

for any $\mathcal{A}$ (with equality for the optimal $\mathcal{A}$). Moreover since $\mathbb{A}$'s declaration has $\boldsymbol{\mu}^t$-probability $P_t$,

$$\widetilde{\mathbb{E}}[E_{t+1} \mid \mathcal{F}_t] \leq \mathbb{E}[E_{t+1} \mid \mathcal{F}_t]/P_t \leq E_t/P_t.$$

$P_t$ is $\mathcal{F}_t$ measurable, so $\widetilde{\mathbb{E}}[P_t E_{t+1} \mid \mathcal{F}_t] \leq E_t$. This establishes the claim and ends the proof. $\qquad\square$

We prove Theorem 3.2 by constructing a Bayesian adversary who slowly degrades the empirical performance of each arm $a_i$. This adversary declares for each batch of samples that the empirical average reward $\hat{p}_i(n_{i,t})$ of arm $i$ will drop by $\Omega\left(\frac{\varrho}{\log(N)}\right)$, at least once the sample size $n_{i,t} \geq N^\varrho$ is large. This degradation schedule ensures that the average reward of any arm is smaller than $\beta$ once it has been sampled $\Omega(N^{1-\varrho})$ times. Moreover it follows from reverse Chernoff estimates that this adversary pays $O\left(\frac{1}{\log^2(N)}\right)$ cost per sample, leading to a failure probability lower bound of $\exp(-O(N/\log^2 N))$ from Lemma 4. As with the upper bound, sharp constants can be tracked with more work and indeed our optimal adversary uses the same function $\theta$ as in Algorithm 3.

**Conclusion** Our aim in this paper was to understand the sample complexity of pure exploration with infinitely many arms. We showed that, surprisingly, the behavior of fixed confidence and fixed budget problems is provably very different. In the former setting, there is a nearly optimal algorithm which precisely balances between sampling enough distinct arms (to estimate the quantile) and obtaining enough samples of a single arm (to be output). In the latter, the optimal algorithm must repeatedly decide whether to continue with the current arm or switch to a fresh one, via a gradually decreasing sequence of rejection thresholds.

Several interesting questions remain. One is that our fixed budget analysis is tailored to the $\delta \to 0$ setting, and does not apply if $\varepsilon, \eta, \delta$ all tend to zero at comparable rates. Hence other behaviors could be present in such parameter regimes. Additionally, a key conceptual feature of infinite-armed bandits is the possibility that no "good" arms are among those sampled by the algorithm. By definition, this simply cannot happen in $K$-armed bandits. It would be interesting to identify a natural problem setting that interpolates between them. Finally high probability bounds on the fixed confidence sample complexity would interpolate between the two settings we studied.

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

# A  Proofs from Section 2

---

**Algorithm 4:** Output $\hat{\alpha} \in \left[G^{-1}(1 - \eta_1) - \frac{\varepsilon}{3}, G^{-1}\left(1 - \eta_1 + \eta_2\right) + \frac{\varepsilon}{3}\right]$ with probability $1 - \frac{\delta}{2}$

---
1 **input:** arm set $\mathcal{S} = (a_1, a_2, \dots)$ and parameters $(\eta_1, \eta_2, \varepsilon, \delta) \in (0, 1)$ with $\eta_2 < \eta_1$.
2 initialize: $K = \frac{C\eta_1 \log(1/\delta)}{\eta_2^2}$.
3 **for** $i = 1, 2, \dots, K$ **do**
4 $\quad$ Collect $n = \frac{C \log(1/\eta_2)}{\varepsilon^2}$ samples of arm $i$. Set $\hat{p}_i = \hat{p}_i(n)$ to be the average observed reward.
5 **end**
6 Let $\hat{\alpha}$ be the $k$-th largest value in $\{\hat{p}_1, \dots, \hat{p}_K\}$ for $k = \left\lceil K\left(\eta_1 - \frac{\eta_2}{2}\right)\right\rceil$.
7 Return $\hat{\alpha}$

---

We show the following generalization of Proposition 2.1.

**Proposition A.1.** *Fix $0 \leq \eta_1, \eta_2, \varepsilon, \delta \leq 1$ with $\eta_2 \leq \eta_1$. With probability at least $1 - \frac{\delta}{2}$, the output $\hat{\alpha}$ of Alg. 4 satisfies*

$$\hat{\alpha} \in \left[G^{-1}(1 - \eta_1) - \frac{\varepsilon}{3}, G^{-1}\left(1 - \eta_1 + \eta_2\right) + \frac{\varepsilon}{3}\right].$$

*Moreover, Alg. 4 has sample complexity*

$$O\left(\frac{\eta_1 \log(1/\eta_2) \log(1/\delta)}{\eta_2^2 \varepsilon^2}\right).$$

*Proof.* The sample complexity is clear so we focus on the first statement. First observe that by a Chernoff estimate, for each $i \in [K]$,

$$\mathbb{P}\left[|p_i - \hat{p}_i| \geq \frac{\varepsilon}{3}\right] \leq \frac{\eta_2}{8}. \tag{A.1}$$

Let $N(\varepsilon)$ be the number of $i \in [K]$ such that $|p_i - \hat{p}_i| \geq \frac{\varepsilon}{3}$. Applying a second Chernoff estimate (of multiplicative form, see e.g. Theorem 4.5 in [MU17]) on these events as $i$ varies and noting that $K\eta_2 \geq C \log(1/\delta)$, (A.1) implies

$$\mathbb{P}\left[N(\varepsilon) \leq \frac{K\eta_2}{6}\right] \geq 1 - \frac{\delta}{8}. \tag{A.2}$$

We next show that with probability at least $1 - \frac{\delta}{4}$,

$$\hat{\alpha} \leq \overline{\alpha} + \frac{\varepsilon}{3} \equiv G^{-1}\left(1 - \eta_1 + \eta_2\right) + \frac{\varepsilon}{3}. \tag{A.3}$$

With $p_i$ the (true) mean reward from arm $a_i$, let

$$N_{\overline{\alpha}} \equiv |\{i \in [K] \ : \ p_i > \overline{\alpha}\}|$$

denote the number of the $K$ tested arms which satisfy $p_i > \overline{\alpha}$. By definition, $N_{\overline{\alpha}}$ is stochastically dominated by a $\text{Bin}\left(K, \eta_1 - \frac{9\eta_2}{10}\right)$ random variable, and $\eta_1 - \frac{3\eta_2}{4} = \Theta(\eta_1)$ since $\eta_2 \leq \eta_1$. Note that

$$\eta_1 - \frac{9\eta_2}{10} \asymp \eta_1 - \frac{3\eta_2}{4} \asymp \eta_1,$$

$$\frac{\eta_1 - \frac{9\eta_2}{10}}{\eta_1 - \frac{3\eta_2}{4}} \geq 1 + \frac{\eta_2}{20\eta_1}.$$

Therefore another multiplicative Chernoff estimate implies

$$\mathbb{P}\left[N_{\overline{\alpha}} \leq K\left(\eta_1 - \frac{3\eta_2}{4}\right)\right] \geq e^{-\Omega(K\eta_2^2/\eta_1)} \geq 1 - \frac{\delta}{8}.$$

When both $N(\varepsilon) \leq \frac{K\eta_2}{6}$ and $N_{\overline{\alpha}} \leq K\left(\eta_1 - \frac{3\eta_2}{4}\right)$ hold, it follows by definition that $\hat{\alpha} \leq \overline{\alpha} + \frac{\varepsilon}{3}$. Hence recalling (A.2) above, we conclude that

$$\mathbb{P}\left[\hat{\alpha} \leq \overline{\alpha} + \frac{\varepsilon}{3}\right] \geq 1 - \frac{\delta}{4},$$

establishing (A.3). The other direction is similar. With $\alpha = G^{-1}(1 - \eta_1)$ as usual, we set

$$N_\alpha \equiv |\{i \in [K] : p_i \geq \alpha\}|. \tag{A.4}$$

This time, $N_\alpha$ stochastically dominates a $\text{Bin}(K, \eta_1)$ random variable. Yet another Chernoff estimate yields

$$\mathbb{P}\left[N_\alpha \geq K\left(\eta_1 - \frac{\eta_2}{4}\right)\right] \geq 1 - \frac{\delta}{8}.$$

Using (A.2) in the same way as above, we find

$$\mathbb{P}\left[\hat{\alpha} \geq \alpha - \frac{\varepsilon}{3}\right] \geq 1 - \frac{\delta}{4}.$$

This concludes the proof. $\qquad\square$

*Proof of Theorem 2.1.* First we analyze the expected sample complexity. On the event that

$$\hat{\alpha} \in \left[G^{-1}(1-\eta) - \frac{\varepsilon}{3}, G^{-1}\left(1 - \frac{\eta}{2}\right) + \frac{\varepsilon}{3}\right] \tag{A.5}$$

we claim that Alg. 2 terminates with probability $\eta/4$ for each $a_i$. Indeed, if

$$\hat{p}_i \geq G^{-1}\left(1 - \frac{\eta}{2}\right)$$

then termination always happens by definition. This has probability at least $1/4$ if $p_i \geq G^{-1}\left(1 - \frac{\eta}{2}\right)$ by Theorem 1 in [GM14], and the latter condition holds with probability at least $\eta/2$ by definition. It follows that when (A.5) holds, the expected sample complexity of Alg. 2 is $O\left(\frac{\log(1/\eta\delta)}{\eta\varepsilon^2}\right)$. On the other hand, (A.5) fails to hold with probability less than $\delta$. Because of the explicit termination condition in Alg. 2, this yields a additional sample complexity contribution of smaller order $O\left(\delta \log(1/\delta)\frac{\log(1/\eta\delta)}{\eta\varepsilon^2}\right)$. Finally Alg. 4 has sample complexity

$$O\left(\frac{\log(1/\eta)\log(1/\delta)}{\eta\varepsilon^2}\right)$$

which clearly forms the dominant contribution. This completes the proof of the sample complexity bound and we now turn to proving correctness with probability $1 - \delta$. First, it is easy to see that Alg. 4 outputs some arm $a_i$ with probability at least $1 - \frac{\delta}{2}$. It therefore suffices to show that for any fixed $\hat{\alpha}$ satisfying (A.5), conditioned on the event $\hat{p}_i \geq \hat{\alpha} - \frac{\varepsilon}{3}$, the conditional probability that $p_i \geq \alpha - \varepsilon$ is at least $1 - \frac{\delta}{2}$.

We do this using Bayes' rule. If $p_i \geq G^{-1}(1 - \frac{\eta}{2})$, then as above Theorem 1 in [GM14] implies

$$\mathbb{P}\left[\hat{p}_i \geq \hat{\alpha} - \frac{\varepsilon}{3}\right] \geq \mathbb{P}[\hat{p}_i \geq p_i] \geq 1/4.$$

This event hence contributes probability at least $\eta/4$ to the event $p_i \geq G^{-1}(1 - \eta)$. On the other hand, if $p_i \leq G^{-1}(1 - \eta) - \varepsilon \leq \hat{\alpha} - \frac{2\varepsilon}{3}$, then

$$\mathbb{P}\left[\hat{p}_i \geq \hat{\alpha} - \frac{\varepsilon}{3}\right] \leq \mathbb{P}\left[\hat{p}_i \geq p_i + \frac{\varepsilon}{3}\right] \leq \eta\delta/8$$

for an absolute constant $C$. Combining these via Bayes' rule implies the desired result. $\qquad\square$

# B   Lower Bound for Fixed Budget

**Fixed Budget with Unknown $\alpha$**

Before giving the proof, we give some qualitative discussion of the role of unknown $\alpha$. We consider Theorem 3.2 to be a definitive lower bound, since e.g. being given the value of $\alpha$ only makes the result stronger. When $\alpha$ is unknown, it is possible to give an essentially matching algorithm, but more care is required when stating the result. This is inherent and stems from the fact that the value $\alpha = G_\mu^{-1}(1 - \eta)$ can be difficult or even impossible to estimate, yet determines the constant $c_{\alpha,\beta}$ in the desired rate.

Let us illustrate the issue by a counterexample. Consider $\mu_N$ defined by:

$$\mathbb{P}^{p\sim\mu_N}[p = 0.4] = \frac{1}{2} + e^{-10N},$$
$$\mathbb{P}^{p\sim\mu_N}[p = 0.2] = \frac{1}{2} - e^{-10N}. \tag{B.1}$$

Similarly define $\tilde{\mu}_N$ by:

$$\mathbb{P}^{p\sim\tilde{\mu}_N}[p = 0.4] = \frac{1}{2} - e^{-10N},$$
$$\mathbb{P}^{p\sim\tilde{\mu}_N}[p = 0.3] = 2e^{-10N}, \tag{B.2}$$
$$\mathbb{P}^{p\sim\tilde{\mu}_N}[p = 0.2] = \frac{1}{2} - e^{-10N}.$$

Then $\mu_N$ and $\tilde{\mu}_N$ are not distinguishable using $N$ samples, yet $G_\mu^{-1}(1/2) = 0.4$ while $G_{\tilde{\mu}}^{-1}(1/2) = 0.3$. Using non-distinguishability it follows that the lower bound of Theorem 3.2 applies to $\tilde{\mu}_N$ with threshold $\alpha = G_{\mu_N}^{-1}(1/2) = 0.4$, as opposed to the direct application using $G_{\tilde{\mu}_N}^{-1}(1/2) = 0.3$. It is not hard to show using monotonicity of $\frac{1}{\sqrt{x(1-x)}}$ that

$$c_{0.4,0.4-\varepsilon} < c_{0.3,0.3-\varepsilon}$$

for all $\varepsilon \leq 0.3$. As a result, it is information-theoretically **impossible** to achieve the rate (3.1) for $\tilde{\mu}_N$ if the target quantile value $\alpha$ is not given. The core reason is that the value $G_{\tilde{\mu}}^{-1}(1/2) = 0.3$ is too sensitive to the choice $\eta = 1/2$ of quantile.

Fortunately, this issue is more of an annoyance than a real difficulty. It can be fixed in several ways. In Theorems B.1, B.2, and B.3 below we give three concrete formulations under which the guarantee (3.1) can be achieved, as mentioned in the main body.

**Theorem B.1.** *For fixed $\eta_1, \eta_2, \varepsilon$, there is a sequence $(\mathcal{A}_N)_{N\geq 1}$ of $N$-sample algorithms outputting $a_{i^*}$ such that the following holds for any sequence $(\mu_N)_{N\geq 1}$ of reservoir distributions. Letting*

$$\alpha_N = \frac{1}{\eta_1 - \eta_2} \cdot \int_{1-\eta_1}^{1-\eta_2} G_{\mu_N}^{-1}(x)dx$$

*be a quantile average of $\mu_N$, we have*

$$\limsup_{N\to\infty} \frac{(-\log\mathbb{P}[p_{i^*} < \alpha_N - \varepsilon])\cdot\log^2 N}{c_{\alpha_N,\alpha_N-\varepsilon}N} \geq 1. \tag{B.3}$$

**Theorem B.2.** *For fixed $\eta, \varepsilon$, there is a sequence $(\mathcal{A}_N)_{N\geq 1}$ of $N$-sample algorithms outputting $a_{i^*}$ such that for any sequence of reservoir distributions $\mu_N$ satisfying*

$$\alpha_N \equiv G_{\mu_N}^{-1}(1-\eta) \geq \frac{1+\varepsilon}{2},$$

*we have*

$$\limsup_{N\to\infty} \frac{(-\log\mathbb{P}[p_{i^*} < G_{\mu_N}^{-1}(1-\eta) - \varepsilon])\cdot\log^2 N}{c_{\alpha_N,\alpha_N-\varepsilon}N} \geq 1. \tag{B.4}$$

**Theorem B.3.** *For any fixed $\varepsilon_1 > \varepsilon$, there is a sequence $(\mathcal{A}_N)_{N\geq 1}$ of $N$-sample algorithms outputting $a_{i^*}$ such that for any fixed reservoir distribution $\mu$ with $\mu^* > \varepsilon$,*

$$\limsup_{N\to\infty} \frac{(-\log\mathbb{P}[p_{i^*} < \mu^* - \varepsilon_1])\cdot\log^2 N}{N} \geq c_{\mu^*,\mu^*-\varepsilon}. \tag{B.5}$$

We emphasize that the rate (3.1) is optimal in all cases since the lower bound of Theorem 3.2 is for an easier problem. The first formulation above may be the most principled choice. The idea is that an averaged quantile depends continuously on $\mu$, and can in fact be estimated by applying Proposition A.1 for several pairs $(\eta_1, \eta_2)$ and computing a Riemann sum. The second formulation requires only the mild condition that $\alpha \geq \frac{1+\varepsilon}{2}$ and uses monotonicity of $c_{\alpha,\alpha-\varepsilon}$ on this set. (In other words, if the average reward values $p$ appearing in (B.1), (B.2) were larger than 0.5, there would be no counterexample.) The third formulation allows us to almost send $\eta$ all the way down to 0. It uses the fact that

$$\mu^* - (\varepsilon_1 - \varepsilon) \leq G_\mu^{-1}(1-\eta')$$

for some $\eta' = \eta'(\mu, \varepsilon_1, \varepsilon) > 0$. These results show that (3.1) is achievable even without knowledge of $\alpha$, up to a choice of technical modification to sidestep the counterexample discussed above.

**Remark B.1.** *In fact uniformity in $(\alpha, \beta)$ holds in the following sense. For any sequence $(\alpha_N, \beta_N)_{N \geq 1}$ of pairs with $\min(\beta_N, \alpha_N - \beta_N, 1 - \alpha_N)$ uniformly bounded below, there is a sequence $(\mathcal{A}_N)_{N \geq 1}$ of $N$-sample algorithms such that for any $\eta \in (0,1)$ and any sequence of reservoir distributions $\mu_N$ with $G_{\mu_N}^{-1}(1 - \eta) \geq \alpha_N$,*

$$\limsup_{N \to \infty} \frac{(-\log \mathbb{P}[p_{i^*} < \beta_N]) \cdot \log^2 N}{c_{\alpha_N, \beta_N} N} \leq 1. \tag{B.6}$$

*This can be shown identically to Theorem 3.1, though we don't give the proof in this generality. It is useful for the reduction arguments in Theorems B.1, B.2, and B.3.*

## B.1 Preparation for the Proof

Here we prove Theorem 3.2. For any $\alpha, \beta, \eta, \varrho > 0$ we construct a reservoir $\mu = \mu_{\alpha, \beta, \eta, \varrho}$ such that

$$\liminf_{N \to \infty} \frac{(-\log \mathbb{P}^\mu[p_{i^*} < \beta]) \cdot \log^2 N}{N} \leq c_{\alpha, \beta} + \lambda(\varrho) \tag{B.7}$$

holds for any sequence of $N$-sample algorithms $\mathcal{A}_N$, and where $\lim_{\varrho \to 0} \lambda(\varrho) = 0$ for fixed $\alpha, \beta, \eta$.

## B.2 Admissible Reservoirs and Bayesian Perspective

In proving Theorem 3.2, we will use reservoir distributions $\mu$ of a specific form. Namely, we require each $\mu$ to be supported on an interval $[\underline{\gamma}, \overline{\gamma}]$, where

$$0 < \beta - \varrho < \underline{\gamma} < \beta < \alpha < \overline{\gamma} < \alpha + \varrho < 1.$$

In fact we define $\underline{\gamma}, \overline{\gamma}$ explicitly (recall that $\varrho > 0$ is a small constant which we eventually send to 0) by

$$\begin{aligned} \theta(\underline{\gamma}) &= \theta(\beta) - \varrho^2; \\ \theta(\overline{\gamma}) &= \theta(\alpha) + \varrho^2. \end{aligned} \tag{B.8}$$

We say $\mu$ is $(\underline{\gamma}, \overline{\gamma}, \underline{f}, \overline{f})$ *admissible* if $\mu$ has density $\mu(dx) = f(x)dx$ for a Borel measurable function $f$ and satisfies for constants $0 < \underline{f} < \overline{f} < \infty$,

$$f(x) \in [\underline{f}, \overline{f}], \quad \forall x \in [\underline{\gamma}, \overline{\gamma}].$$

Towards proving Theorem 3.2, we fix throughout this section some $(\underline{\gamma}, \overline{\gamma}, \underline{f}, \overline{f})$ admissible $\mu$ such that $G_\mu^{-1}(\alpha) = \eta$ holds, for appropriate constants $(\underline{f}, \overline{f})$ depending only on $(\eta, \varepsilon, \alpha, \beta, \underline{\gamma}, \overline{\gamma})$. It is easy to see that this is always possible.

An admissible $\mu$ is roughly comparable to the uniform distribution on an interval. Using admissible reservoirs gives each $a_i$ the potential to slowly degrade in observed quality over time. We remark that while it is more convenient to work with reservoirs supported away from the boundaries, i.e. in $[\underline{\gamma}, \overline{\gamma}] \subseteq (0,1)$, we do not expect this to be essential.

It will be helpful throughout this section to take a Bayesian point of view. We treat $\mu_N$ as known to $\mathcal{A}_N$, since $\mathcal{A}_N$ is in fact allowed to depend on $\mu_N$. Thus at each time $t$, each $p_i$ has a posterior probability distribution which we denote by $\mu_{i,t}$. Note that each $\mu_{i,t}$ depends only on $(n_{i,t}, \hat{p}_{i,t})$ and is initialized at $\mu_{i,0} = \mu$. We denote by

$$\boldsymbol{\mu}^t = (\mu_{1,t}, \mu_{2,t}, \dots) \tag{B.9}$$

the sequence of posterior distributions $\mu_{i,t}$. Since arms are independent, $\boldsymbol{\mu}^t$ is the full time-$t$ posterior of the algorithm.

## B.3 Batched Algorithms and Adversaries

In pure exploration problems, it is possible to significantly simplify the structure of any algorithm at the cost of a small multiplicative increase in the sample complexity. We carry this out using the notion of a batch-compressed algorithm.

**Definition B.1.** *Given an increasing sequence $B = (b_1, b_2, \dots)$ of positive integers, an algorithm $\mathcal{A}$ is B-**batch-compressed** if $\mathcal{A}$ can only act by increasing the number of times $n_i$ that $a_i$ has been sampled from $b_k$ to $b_{k+1}$, so that $n_i \in B$ holds at all times. $B$ is $\varrho$-**slowly increasing** if*

$$\frac{b_{k+1}}{b_k + 1} \leq 1 + \varrho, \quad \forall k \geq 1.$$

*Finally if $\mathcal{A}$ is B-batch-compressed and $B$ is $\varrho$-slowly increasing, we say that $\mathcal{A}$ is $\varrho$-batch-compressed.*

Unlike the batched algorithms studied in [PRCS16, GHRZ19], batch-compression is only important for us as an analysis technique. Indeed the following proposition shows that it does not fundamentally affect pure exploration algorithms.

**Proposition B.2.** *If $B$ is $\varrho$-slowly increasing, then for any $N$-sample algorithm $\mathcal{A}$, there exists an $B$-batch-compressed $\lfloor N(1 + \varrho) \rfloor$ algorithm $\mathcal{A}'$ with the same output.*

*Proof.* We show how to simulate $\mathcal{A}$ using the $B$-batch-compressed $\mathcal{A}'$, assuming that the sequence of rewards for each $a_i$ is fixed. Each time $\mathcal{A}$ samples arm $i$ for the $n_i = (a_k + 1)$-st time for $a_k \in A$, $\mathcal{A}'$ samples arm $i$ until $n_i = a_{k+1}$. Then $\mathcal{A}'$ has all the information of $\mathcal{A}$ at all times, hence can simulate the behavior and output of $\mathcal{A}$. Moreover by the definition of $\varrho$-slowly increasing, the sample complexity of $\mathcal{A}'$ is larger than that of $\mathcal{A}$ by at most a factor $(1 + \varrho)$. $\qquad\square$

We will use the above with $\varrho \to 0$ slowly as $N \to \infty$. Then the sample complexity increase $1 + \varrho$ is absorbed into the $1 + o(1)$ factor in Theorem 3.2. As a result it suffices to establish (B.7) under the additional assumption that $\mathcal{A}_N$ is $\varrho$-batch-compressed.

## B.4 Fisher Information Distance

Determining the tight constant $c_{\alpha,\beta}$ requires significant care. In particular the adversary must decrease the empirical average rewards $\hat{p}_{i,t}$ at a precise rate depending on $n_{i,t}$. This rate turns out to involve the *Fisher information distance*. For $a, b \in [0, 1]$ we define the Fisher information distance $d_F(a, b)$ between $a$ and $b$ to be

$$d_F(a, b) = \left| \int_a^b \frac{dx}{\sqrt{x(1-x)}} \right|.$$

This agrees with the more general Fisher information metric when each $a \in [0, 1]$ is identified with the corresponding Bernoulli distribution. We refer the reader to [Nie20] for a survey on information geometry. In short, the Fisher information yields a natural Riemannian metric on families of probability distributions which are parametrized by smooth manifolds. However we will use only elementary properties of $d_F$.

We parametrize $[0, 1]$ using the function $\theta : [0, 1] \to [0, \pi]$ defined by

$$\theta(a) = d_F(0, a) = \int_0^a \frac{dx}{\sqrt{x(1-x)}} = \arccos(1 - 2a). \tag{B.10}$$

In particular,

$$d_F(a, b) = |\arccos(1 - 2a) - \arccos(1 - 2b)| \geq 2|a - b|$$

and so $d_F(0, 1) = \pi$. The main property of $\theta$ that we will use is the resulting differential equation

$$\theta'(a) = \frac{1}{\sqrt{\theta(a)(1 - \theta(a))}}. \tag{B.11}$$

In our case, $\theta^{-1}$ parametrizes a "constant speed" path through the space of Bernoulli variables, viewing the Fisher information. Correspondingly, our adversary will ensure that $\theta(\hat{p}_i(n_{i,t}))$ decreases linearly in $\log(n_{i,t})$.

### B.5 Preliminary Lemmas from Moderate Deviations

Recall that for positive integers $a$ and $b$, the $\mathrm{Beta}(a, b)$ distribution has probability density function

$$\frac{(a+b-1)!}{(a-1)!(b-1)!}x^{a-1}(1-x)^{b-1}$$

for $x \in [0, 1]$. We now recall a moderate deviations principle for the binomial distribution and a central limit theorem for the beta distribution.

**Lemma 4** (Theorem 2.2 in [DA92]). *For any $0 < \underline{q} < \overline{q} < 1$ and constant $\varrho > 0$ there exists $\Delta_0(\underline{q}, \overline{q}, \varrho)$ and $M_0(\underline{q}, \overline{q}, \varrho)$ such that the following holds for all $p \in [\underline{q}, \overline{q}]$. For $n \geq n_0(\underline{q}, \overline{q}, \varrho)$ sufficiently large and any $\frac{1}{\Delta_0\sqrt{n}} \leq \Delta \leq \Delta_0$ we have*

$$e^{\left(-\frac{\Delta^2}{2p(1-p)}-\varrho\right)n} \leq \mathbb{P}\left[\frac{Bin(n,p)}{n} \leq p - \delta\right] \leq e^{\left(-\frac{\Delta^2}{2p(1-p)}+\varrho\right)n}.$$

**Lemma 5** (Lemma A.1 in [MNS16]). *Let $\{a_n\}_{n\geq n_0}$ be a sequence satisfying*

$$\underline{\gamma} \leq \frac{a_n}{n} \leq \overline{\gamma}.$$

*Then the $\mathrm{Beta}(n - a_n + 1, a_n + 1)$ distribution on $[0, 1]$ obeys a central limit theorem with mean $\frac{a_n}{n}$ and standard deviation $\sqrt{\frac{(a_n/n)(1-(a_n/n))}{n}}$ in the sense that for any bounded sequence $(w_n)_{n\geq n_0}$ of real numbers and with $\Phi$ the normal CDF,*

$$\lim_{n\to\infty}\left|\Phi(w_n) - \mathbb{P}^{x\sim\mathrm{Beta}(n-a_n+1,a_n+1)}\left[(x - (a_n/n)) \cdot \sqrt{\frac{n}{(a_n/n)(1-(a_n/n))}} \leq w_n\right]\right| = 0.$$

In the next two lemmas, we lower bound the probability that $\hat{p}_{i,t}$ changes significantly when the number $n_{i,t}$ of samples for $a_i$ increases by a factor $(1 + \varrho)$.

**Lemma 6.** *Assume $\mu$ is $(\underline{\gamma}, \overline{\gamma}, \underline{f}, \overline{f})$-admissible. Suppose that arm $i$'s average reward $\hat{p}_{i,t}$ after $n = n_{i,t}$ samples satisfies*

$$\hat{p}_{i,t} \in [\beta, \overline{\gamma}]. \tag{B.12}$$

*Then for $n \geq C(\underline{\gamma}, \overline{\gamma}, \underline{f}, \overline{f}, \beta)$ sufficiently large,*

$$\mathbb{P}^{x\sim\mu_{i,n}}\left[x \leq \hat{p}_{i,t}\right] \geq \frac{\underline{f}}{3\overline{f}}. \tag{B.13}$$

*Proof.* Let $R_{i,t} = n\hat{p}_{i,t}$ be the total reward from arm $i$ so far. The posterior distribution $\mu_{i,t}$ for $p_i$ takes the form

$$\mu_{i,t}(dx) = \frac{x^{R_{i,t}}(1-x)^{n-R_{i,t}}f(x)dx}{\int_{\underline{\gamma}}^{\overline{\gamma}} x^{R_{i,t}}(1-x)^{n-R_{i,t}}f(x)dx}.$$

For $x \in [\underline{\gamma}, \overline{\gamma}]$ we estimate

$$\frac{x^{R_{i,t}}(1-x)^{n-R_{i,t}}f(x)}{\int_{\underline{\gamma}}^{\overline{\gamma}} x^{R_{i,t}}(1-x)^{n-R_{i,t}}f(x)dx} \geq (\underline{f}/\overline{f}) \cdot \frac{x^{R_{i,t}}(1-x)^{n-R_{i,t}}}{\int_0^1 x^{R_{i,t}}(1-x)^{n-R_{i,t}}dx}.$$

The right-hand side is the density of a beta variable with parameters $(R_{i,t} + 1, n - R_{i,t} + 1)$. We conclude that

$$\mathbb{P}^{x\sim\mu_{i,t}}\left[x \in [\underline{\gamma}, \hat{p}_{i,t}]\right] \geq (\underline{f}/\overline{f}) \cdot \mathbb{P}^{z\sim\mathrm{Beta}(n-R_{i,t}+1,R_{i,t}+1)}\left[z \in [\underline{\gamma}, \hat{p}_{i,t}]\right]$$

For $n$ sufficiently large, it follows from Lemma 5 and (B.12) that

$$\mathbb{P}^{z\sim\mathrm{Beta}(n-R_{i,t}+1,R_{i,t}+1)}\left[z \in [\underline{\gamma}, \hat{p}_{i,t}]\right] \geq \frac{1}{3}.$$

Therefore $\mathbb{P}^{\mu_{i,t}}[p_i \leq \hat{p}_{i,t}] \geq \frac{1}{3}$, proving (B.13). $\square$

**Lemma 7.** *Assume $\mu$ is $(\underline{\gamma}, \overline{\gamma}, \underline{f}, \overline{f})$-admissible and that* (B.12) *holds. For $n = n_{i,t}$, let $\tilde{n} \geq 1$ satisfy $|\tilde{n} - \varrho n| \leq 2$. Let*

$$\tilde{p}_i = \frac{R_{i,n+\tilde{n}} - R_{i,n}}{\tilde{n}}$$

*be the average reward from the $(n+1)$-th through $(n+\tilde{n})$-th samples of arm $i$. Then as $n \to \infty$, for any sequence $\Delta_n = \Theta(1/\log n)$,*

$$\mathbb{P}^t[\tilde{p}_i \leq \theta^{-1}(\theta(\hat{p}_{i,t}) - \Delta_n)] \geq \exp\left(-\frac{n\varrho\Delta_n^2(1 + o_n(1))}{2}\right). \tag{B.14}$$

*Proof.* Stochastic monotonicity implies that

$$\mathbb{P}\left[\frac{\mathrm{Bin}(\tilde{n}, p)}{\tilde{n}} \leq \theta^{-1}\big(\theta(\hat{p}_{i,t}) - \Delta_n\big)\right]$$

is a decreasing function of $p \in [0, 1]$. Combining with Lemma 6, it follows that

$$\mathbb{P}^t[\tilde{p}_i \leq \theta^{-1}(\theta(\hat{p}_{i,t}) - \Delta_n)] = \int \mathbb{P}\left[\frac{\mathrm{Bin}(\tilde{n}, x)}{\tilde{n}} \leq \theta^{-1}\big(\theta(\hat{p}_{i,t}) - \Delta_n\big)\right] d\mu_{i,t}(x)$$

$$\geq \mathbb{P}^{\mu_{i,t}}[p_i \leq \hat{p}_{i,t}] \cdot \mathbb{P}\left[\frac{\mathrm{Bin}(\tilde{n}, \hat{p}_{i,t})}{\tilde{n}} \leq \theta^{-1}\big(\theta(\hat{p}_{i,t}) - \Delta_n\big)\right]$$

$$\geq \frac{\underline{f}}{3\overline{f}} \cdot \mathbb{P}\left[\frac{\mathrm{Bin}(\tilde{n}, \hat{p}_{i,t})}{\tilde{n}} \leq \theta^{-1}\big(\theta(\hat{p}_{i,t}) - \Delta_n\big)\right].$$

Since $\theta$ is smooth with smooth inverse on $[\underline{\gamma}, \overline{\gamma}]$ and $\Delta_n \leq o_n(1)$, we have

$$\hat{p}_{i,t} - \theta^{-1}\big(\theta(\hat{p}_{i,t}) - \Delta_n\big) = (1 \pm o_n(1))\Delta_n \cdot (\theta^{-1})'\big(\theta(\hat{p}_{i,t})\big)$$

$$= \frac{(1 \pm o_n(1)) \cdot \Delta_n}{\theta'(\theta^{-1}(\hat{p}_{i,t}))}$$

$$= (1 \pm o_n(1)) \cdot \Delta_n\sqrt{\hat{p}_{i,t}(1 - \hat{p}_{i,t})}.$$

The result now follows from Lemma 4, where we absorb the factor $\underline{f}/(3\overline{f})$ into the $o_n(1)$. $\square$

## B.6 Proof of Theorem 3.2

Recall the definition (B.8) of $\underline{\gamma}$ and $\overline{\gamma}$. We require $\mathcal{A}$ to be $B$-batch-compressed for $B = B(N, \varrho)$ containing:

1. All positive integers at most $N^{2\varrho}$.
2. All positive multiples of $\lfloor N^\varrho \rfloor$ at most $N^{6\varrho}$.
3. Integers of the form $\lfloor N^{6\varrho}(1 + \varrho)^j \rfloor$ for $j \geq 0$.

It is easy to see that $B$ thus defined is $\varrho$-slowly increasing for any $\varrho > 0$ and $N$ sufficiently large. We denote $b_k = \lfloor N^{6\varrho}(1 + \varrho)^k \rfloor$ so that $|b_{k+1} - (1 + \varrho)b_k| \leq 2$. (This choice of indexing differs from that of Definition B.1, which will not be used in the sequel.)

We next construct our randomness distorting adversary $\mathbb{A} = \mathbb{A}(N, \varrho)$. For each arm $i$, the adversary $\mathbb{A}$ acts as follows depending on the current number of samples $n_{i,t}$.

1. If $n_{i,t} \leq N^{2\varrho}$, then $\mathbb{A}$ does nothing.
2. When $N^{2\varrho} \leq n_{i,t} < N^{6\varrho}$ increases by $N^\varrho$, $\mathbb{A}$ declares that the average reward of this batch of $N^\varrho$ samples is at most $\overline{\gamma} - N^{-\varrho}$.
3. When $n_{i,t}$ increases from $b_k \geq N^{6\varrho}$ to $b_{k+1}$:
   (a) If $\hat{p}_i(b_k) > \beta$ holds, then $\mathbb{A}$ declares that

$$\theta(\hat{p}_i(b_{k+1})) \leq \theta(\hat{p}_i(b_k)) - \frac{\varrho(1 + 10\varrho)d_F(\alpha, \beta)}{\log N}. \tag{B.15}$$

(b) If $\hat{p}_i(b_k) \leq \beta$ holds, then $\mathbb{A}$ declares that
$$\hat{p}_i(b_{k+1}) \leq \beta.$$

4. When the $\mathcal{A}$ chooses the arm $a_{i*}$ to output, $\mathbb{A}$ declares that $p_{i*} < \beta$.

Due to step 4, the declarations made by $\mathbb{A}$ ensure that $p_{i*} < \beta$. Recalling Lemma 4 and Proposition B.2, it remains to show the upper bound

$$\text{strength}(\mathbb{A}) \leq \frac{(c_{\alpha,\beta} + C_*\varrho)N}{\log^2(N)}$$

for a constant $C_* = C_*(\underline{\gamma}, \overline{\gamma}, \underline{f}, \overline{f}, \beta, \alpha)$ independent of $\varrho$ (and $N$). We show this bound in several parts. Recalling (3.4), we refer to the *cost* of a step above as the contribution to Cost from the corresponding declarations by $\mathbb{A}$. The most important parts are Lemmas 10 and 11, which bound the cost of the main step 3a and form the dominant contribution to Cost. Note that throughout the analysis below, all cost upper bounds hold almost surely and we **assume that all of $\mathbb{A}$'s declarations hold true**.

**Lemma 8.** *The total cost from step 2 is at most $C_* N^{1-\varrho}$, for $N \geq C(\underline{\gamma}, \overline{\gamma}, \underline{f}, \overline{f}, \beta, \alpha, \varrho)$ sufficiently large.*

*Proof.* The probability for each such declaration by $\mathbb{A}$ is at least

$$\mathbb{P}[\text{Bin}(N^{2\varrho}, \overline{\gamma}) \leq \overline{\gamma} N^{2\varrho} - N^{\varrho}] \tag{B.16}$$

since $p_i \leq \overline{\gamma}$ almost surely. Recall that a $\text{Bin}(N^{2\varrho}, \overline{\gamma})$ random variable obeys a central limit theorem centered at $\overline{\gamma} N^{2\varrho}$ with standard deviation at least $C(\overline{\gamma}) N^{\varrho}$. Therefore the probability in (B.16) is at least $\frac{1}{3}$ for $N$ is sufficiently large depending on $\varrho$. Hence each such declaration costs at most $C_*$ for $N$ sufficiently large. Moreover such declarations can occur only $N^{1-\varrho}$ times because each one involves $N^{\varrho}$ samples, and the base algorithm $\mathcal{A}$ is an $N$-sample algorithm. This completes the proof. $\square$

**Lemma 9.** *The total cost from step 3b is at most $C_* N^{1-6\varrho}$ as long as $N \geq C(\underline{\gamma}, \overline{\gamma}, \underline{f}, \overline{f}, \varrho)$.*

*Proof.* It suffices to show that the cost per step 3b declaration is at most $C_*$. This follows from (B.13) and stochastic monotonicity. $\square$

**Lemma 10.** *The total cost from step 3a is at most*

$$\frac{N}{\log^2(N)} \cdot (c_{\alpha,\beta} + C_*\varrho + o_N(1)).$$

*Proof.* We claim that the cost from a single instance of step 3a when increasing from $b_k$ to $b_{k+1}$ samples is at most

$$\left(\frac{(b_{k+1} - b_k)}{\log^2(N)}\right)(c_{\alpha,\beta} + C_*\varrho + o_N(1)).$$

This implies the desired result since $\mathcal{A}_N$ is an $N$-sample algorithm. Taking $\Delta = (1 + 10\varrho)d_F(\alpha, \beta)/\log(N)$ in Lemma 7, we find that the declared event has probability at least

$$\exp\left(-\frac{(b_{k+1} - b_k)(1 + 10\varrho)^2 d_F(\alpha, \beta)^2 (1 + o_N(1))}{2\log^2(N)}\right) \geq \exp\left(-\frac{(b_{k+1} - b_k)}{\log^2(N)}(c_{\alpha,\beta} + C_*\varrho + o_N(1))\right).$$

This implies the desired claim and completes the proof. $\square$

**Lemma 11.** *For any $a_i$ sampled $b_0 = \lfloor N^{6\varrho} \rfloor$ times, $\hat{p}_i(b_0) \leq \overline{\gamma}$.*

*Proof.* By definition of $\mathbb{A}$,

$$\begin{aligned}
\hat{p}_i(b_0) &\leq \frac{N^{2\varrho} + (N^{6\varrho} - N^{2\varrho})(\overline{\gamma} - N^{-\varrho})}{N^{6\varrho}} \\
&= \overline{\gamma} - \frac{1}{N^{\varrho}} + \frac{(1 - \overline{\gamma})}{N^{4\varrho}} + \frac{1}{N^{5\varrho}} \\
&\leq \overline{\gamma}.
\end{aligned}$$

In the last step we used the fact that

$$\frac{1}{N^\varrho} \geq \frac{(1-\overline{\gamma})}{N^{4\varrho}} + \frac{1}{N^{5\varrho}}$$

for any $\varrho > 0$ if $N$ is sufficiently large. $\qquad\square$

**Lemma 12.** *For $\varrho \in (0, 1/100)$, if $n_{i,t} \geq N^{1-\varrho}$ and the declarations of $\mathbb{A}$ hold, then $\hat{p}_{i,t} \leq \beta$.*

*Proof.* We analyze the rate at which the adversary forces $\theta(\hat{p}_i(b_k))$ to decrease. From (B.15) and (11) it follows that for $k$ with $b_k \geq N^{1-\varrho}$, we have

$$\theta(\hat{p}_i(b_k)) \leq \theta(\overline{\gamma}) - \frac{\varrho(1+10\varrho)d_F(\alpha,\beta)\log_{1+\varrho}(N^{1-8\varrho})}{\log N}$$

$$= \theta(\overline{\gamma}) - \frac{\varrho(1+10\varrho)(1-8\varrho)d_F(\alpha,\beta)}{\log(1+\varrho)}$$

$$\leq \theta(\overline{\gamma}) - (1+\varrho)d_F(\alpha,\beta)$$

$$\overset{\text{(B.8)}}{<} \theta(\beta).$$

Here we used the fact that $\log(1+\varrho) \leq \varrho$ and $(1+10\varrho)(1-8\varrho) \geq 1$ for $\varrho \in (0, 1/100)$. Since $\theta$ is increasing, this shows that $\hat{p}_{i,t} = \hat{p}_i(b_k) < \beta$ for $b_k \geq N^{1-\varrho}$, completing the proof. $\qquad\square$

**Lemma 13.** *The cost from step 4 is at most $C_*\big(N^{1-\varrho} + 1\big)$.*

*Proof.* First, if $\hat{p}_{i^*,N} \leq \beta$ then the cost from step 4 is at most $C_*$. On the other hand if $\hat{p}_{i^*,N} > \beta$, then Lemma 11 implies $n_{i^*,N} \leq N^{1-\varrho}$. Since the prior $\mu$ is supported in $[\underline{\gamma}, \overline{\gamma}]$, the likelihood ratio of updates from $N^{1-\varrho}$ samples is almost surely bounded by $e^{C_* N^{1-\varrho}}$. Therefore

$$\mathbb{P}^{x\sim\mu_{i,N}}[x < \beta] \geq e^{-C_* N^{1-\varrho}}\mathbb{P}^{x\sim\mu}[x < \beta]$$

$$\geq e^{-C_* N^{1-\varrho}}\frac{(\beta-\underline{\gamma})\underline{f}}{\overline{f}}.$$

This completes the proof. $\qquad\square$

We now combine the lemmas above to conclude Theorem 3.1 via (B.7).

*Proof of Theorem 3.1.* Let $C'_*$ be a larger constant depending on the same parameters. Then by Lemmas 8, 9, and 13, the total cost from Steps 2, 3b, 4 combines to $C'_* N^{1-\varrho}) \leq o_N(N/\log^2 N)$. The main cost contribution of

$$\frac{N}{\log^2 N}(c_{\alpha,\beta} + C_*\varrho + o_N(1)).$$

comes from Lemma 10, and all other terms are of strictly smaller order. We have thus constructed a reservoir sequence $(\mu_N(\varrho))_{N\geq 1}$ satisfying (B.7) for arbitrary $\varrho > 0$, completing the proof. $\qquad\square$

# C   An Optimal Algorithm with Fixed Budget

Here we provide an asymptotically optimal algorithm which establishes Theorems B.1, B.2, and B.3. In the next subsection in which we show how to reduce the other results mentioned to Theorem 3.1 (in which $\alpha$ is given) using Proposition A.1. Our main focus will then be to prove Theorem 3.1.

We will fix $\varrho > 0$ small and construct a sequence of $N$-sample algorithms $(\mathcal{A}(N, \varrho))$ satisfying the slightly relaxed guarantee

$$\liminf_{N\to\infty} \frac{(-\log(\mathbb{P}^{\mu_N(\varrho)}[p_{i^*} < \beta])) \cdot \log^2 N}{N} \geq c_{\alpha,\beta} - \lambda(\varrho) \qquad (\text{C.1})$$

for a (possibly different) function $\lambda$ satisfying $\lim_{\varrho\to 0} \lambda(\varrho) = 0$ (for fixed $\alpha, \beta, \eta$). Here $(\mu_N)_{N\geq 1}$ is any sequence of reservoir distributions satisfying $G_{\mu_N}^{-1}(1-\eta) = \alpha$. An elementary diagonalization argument then implies Theorem 3.1. Thus it suffices to construct algorithms satisfying (C.1) for any desired $\varrho > 0$.

## C.1 Reduction to Known $\alpha$

We explain why Theorems B.1, B.2, and B.3 all follow from Theorem 3.1 (more precisely, the uniform statement given in Remark B.1). We begin with Theorem B.1, where

$$\alpha_N = \frac{1}{\eta_1 - \eta_2} \cdot \int_{1-\eta_1}^{1-\eta_2} G_{\mu_N}^{-1}(x)dx.$$

Let $J = \lceil \frac{6}{\varepsilon(\eta_1 - \eta_2)} \rceil$ and define

$$\eta^{(j)} = \frac{(J-j)\eta_1 + j\eta_2}{J}, \quad j \in [J].$$

It is easy to see that $\eta^{(j+1)} - \eta^{(j)} \leq \eta^{(j)}$ for all $j$. We next apply Alg. 4 on $(\eta^{(j)}, \eta^{(j+1)} - \eta^{(j)}, \varepsilon', \delta')$ for $0 \leq j \leq J - 1$, with:

$$\varepsilon' = \log^{-1/3}(N),$$
$$\delta' = e^{-\frac{10N}{\log^2(N)}} / J.$$

This requires sample complexity

$$N_A \leq \frac{C(\eta_1, \eta_2)N \log\log(N)}{\log(N)} \leq o_N(N). \tag{C.2}$$

Let $\hat{\alpha}_j$ be the resulting output. With probability $1 - J\delta$, we have for each $0 \leq j \leq J - 1$,

$$\hat{\alpha}_j \in \left[ G^{-1}(1 - \eta^{(j)}) - \frac{\varepsilon}{3}, G^{-1}\left(1 - \eta^{(j+1)}\right) + \frac{\varepsilon}{3} \right]. \tag{C.3}$$

Note that the function $G_\mu^{-1}$ is increasing and $[0,1]$-valued. Therefore if (C.3) holds for each $j$, then

$$\left| \frac{1}{J} \cdot \sum_{j=0}^{J-1} \hat{\alpha}_j - \frac{1}{\eta_1 - \eta_2} \cdot \int_{1-\eta_1}^{1-\eta_2} G_{\mu_N}^{-1}(x)dx \right| \leq \frac{\varepsilon}{3} + \frac{1}{J} \leq \frac{\varepsilon}{2}.$$

Therefore the estimator

$$\hat{\alpha}_A = \frac{1}{J} \cdot \sum_{j=0}^{J-1} \hat{\alpha}_j$$

satisfies

$$\mathbb{P}\left[ \left| \hat{\alpha}_A - \frac{1}{\eta_1 - \eta_2} \cdot \int_{1-\eta_1}^{1-\eta_2} G_{\mu_N}^{-1}(x)dx \right| \leq \varepsilon/2 \right] \geq 1 - J\delta' = 1 - e^{-\frac{10N}{\log^2(N)}}.$$

Finally, $c_{\alpha, \alpha - \varepsilon} \leq \pi < 10$ for any $\alpha, \varepsilon \in [0,1]$ (see (B.10)). Therefore the $\delta' = e^{-\frac{10N}{\log^2(N)}}$ failure probability above has a negligible contribution in Theorem B.1. It follows that applying Theorem 3.1 with $\alpha = \hat{\alpha}_A$ as above and $N' = N - N_A$ implies Theorem B.1.

We now turn to Theorem B.2, where $\mu_N$ is required to satisfy $G_{\mu_N}^{-1}(1 - \eta) \geq \frac{1+\varepsilon}{2}$. We run Alg. 4 with parameters

$$\eta_1 = \eta,$$
$$\eta_2 = \log^{-1/3}(N),$$
$$\varepsilon' = \log^{-1/3}(N),$$
$$\delta' = e^{-\frac{10N}{\log^2(N)}}.$$

The sample complexity $N_B$ again satisfies $N_B \leq o(N)$ exactly as in (C.2). Let $\hat{\alpha}_B + \varepsilon'$ be the resulting output. Then with probability at least $1 - e^{-\frac{10N}{\log^2(N)}}$,

$$\hat{\alpha}_B \geq G_{\mu_N}^{-1}(1 - \eta) - 2\varepsilon'$$

and so with $\varepsilon'' = \varepsilon - 2\varepsilon'$, we have

$$\hat{\alpha}_B - \varepsilon'' \geq G_{\mu_N}^{-1}(1 - \eta) - \varepsilon.$$

Moreover, also with probability at least $1 - e^{-\frac{10N}{\log^2(N)}}$,

$$\hat{\alpha}_B \leq G_{\mu_N}^{-1}(1 - \eta + \eta_2).$$

It follows that applying the algorithm of Theorem 3.1 with

$$(N, \alpha, \eta, \varepsilon) = (N - N_B, \hat{\alpha}_B, \eta - \eta_2, \varepsilon - 2\varepsilon')$$

suffices to recover Theorem B.2, since $\eta_2$ and $\varepsilon'$ tend to $0$ as $N \to \infty$. As in our discussion of Theorem B.1 above, the failure probability $e^{-\frac{10N}{\log^2(N)}}$ is negligible compared to the relevant rate in Theorem B.2.

Finally, Theorem B.3 relies on the simple fact

$$\lim_{\eta \to 0} G_\mu^{-1}(1 - \eta) = \mu^{\cdot} \tag{C.4}$$

Recall that $\mu^* \in [0, 1]$ denotes the maximum value in the support of $\mu$. We run Alg. 4 on $(\eta_1, \eta_2, \varepsilon', \delta')$ where:

$$\eta_1 = \log^{-1/3}(N),$$
$$\eta_2 = \eta_1/2,$$
$$\varepsilon' = \varepsilon_1 - \varepsilon,$$
$$\delta' = e^{-\frac{10N}{\log^2(N)}}.$$

It follows from Proposition A.1 that the resulting output $\hat{\alpha}_C + \frac{\varepsilon_1 - \varepsilon}{2}$ is computed using $O\left(\frac{N \log\log(N)}{\log(N)}\right) \leq o(N)$ samples as in the previous cases. Moreover for $N$ sufficiently large:

$$\mathbb{P}\left[\hat{\alpha}_C + \frac{\varepsilon_1 - \varepsilon}{2} \geq \mu^* - \frac{\varepsilon'}{3} - o_N(1)\right] \overset{(C.4)}{\geq} \mathbb{P}\left[\hat{\alpha}_C + \frac{\varepsilon_1 - \varepsilon}{2} \geq G_\mu^{-1}(1 - \eta_1) - \frac{\varepsilon'}{3}\right]$$
$$\geq 1 - \delta'$$
$$= 1 - e^{-\frac{10N}{\log^2(N)}}.$$

Since $\varepsilon_1 > \varepsilon$, this means for $N \geq N_0(\mu, c', \dots)$ large enough,

$$\mathbb{P}\left[\hat{\alpha}_C \geq \mu^* - (\varepsilon_1 - \varepsilon)\right] \geq 1 - e^{-\frac{10N}{\log^2(N)}}.$$

Note that Alg. 4 also ensures that with probability $1 - e^{-\frac{10N}{\log^2(N)}}$,

$$\hat{\alpha}_C \leq \mu^* + \frac{\varepsilon'}{3} - \frac{\varepsilon_1 - \varepsilon}{2} = \mu^* - \frac{\varepsilon_1 - \varepsilon}{6}$$
$$\leq G_\mu^{-1}(1 - \eta')$$

for some $\eta'(\mu, \varepsilon_1, \varepsilon) > 0$. It follows that applying Theorem 3.1 with

$$(N, \alpha, \eta, \varepsilon) = (N - N', \hat{\alpha}_C, \eta', \varepsilon)$$

implies Theorem B.3.

## C.2 The Fixed Budget Algorithm

We now present Algorithm 3 for the fixed budget problem (recall the informal discussion in Section 3). Algorithm 3 studies one arm $a_i$ at a time, moving to $a_{i+1}$ if $a_i$ is rejected. Similarly to the previous section, some details are needed while $n_{t,i}$ is small, since large deviation asymptotics may not have kicked in yet. As explained at the start of the section, we choose a small constant $\varrho > 0$. In fact, we will eventually choose small constants

$$0 < \varrho \ll \varrho_1 \ll \varrho_2 \ll \varrho_3 \ll \varrho_4 \ll \varrho_5 \ll 1$$

which all tend to 0 as $\varrho \to 0$. These constants will be defined throughout the proof. More formally, these values can be obtained by choosing $\varrho_5 > 0$ arbitrarily small, then $\varrho_4 > 0$ sufficiently small depending on $\varrho_5$, and so on.

Algorithm 3 operates in a batch-compressed way, for a sequence $(b_1, b_2, \dots)$ defined as follows:

$$b_0 = \lceil \varrho_1 \log^2(N) \rceil,$$
$$k_0 = \lceil \log_{1+\varrho}\left(\log^4(N)/b_0\right) \rceil$$
$$b_k = b_0(1+\varrho)^k, \quad k \le k_0$$
$$b_{k_0+j} = \lceil (1+\varrho)^j b_{k_0} \rceil, \quad j \ge 1$$
$$\tau_k = \alpha - \varrho - \frac{k}{\sqrt{\log N}}, \quad k \le k_0$$
$$\tau_{k_0+j} = \theta(\alpha - 2\varrho) - j \cdot \frac{d_F(\alpha, \beta)\varrho(1-\varrho_2)}{\log N}, \quad j \ge 1.$$

Note in particular that $b_{k_0} \ge \log^4(N)$. We denote by $\hat{p}_{i,t}$ the empirical average reward collected by $a_i$ from its first $t$ samples.

---

**Algorithm 5:** Output arm with $p_i \ge \beta$ using $N$ samples with high probability

---
1 **input:** an infinite sequence of arms $i = 1, 2, \dots$
2 initialize: $i = 0$
3 **while** *fewer than $N$ samples have been collected* **do**
4    $i \leftarrow i + 1$
5    Collect $b_0$ samples of arm $i$.
6    **if** $\hat{p}_{i,b_0} \le \alpha - \varrho$ **then**
7      **Reject** arm $i$
8    **end**
9    **for** $k = 1, 2, \dots, k_0$ **do**
10      Collect $b_k - b_{k-1}$ samples of arm $i$ for a total of $b_k$ samples.
11      **if** $\hat{p}_{i,b_k} \le \alpha - \varrho - \frac{k}{\sqrt{\log N}}$ **then**
12        **Reject** arm $i$;
13      **end**
14    **end**
15    **for** $j = 1, 2, \dots$ **do**
16      Collect $b_{k_0+j} - b_{k_0+j-1}$ samples of arm $i$ for a total of $b_{k_0+j}$.
17      **if** $\theta(\hat{p}_{i,b_{k_0+j}}) \le \theta(\alpha - 2\varrho) - j \cdot \frac{d_F(\alpha,\beta)\varrho(1-\varrho_2)}{\log N}$ **then**
18        **Reject** arm $i$
19      **end**
20    **end**
21 **end**
22 Return arm $i$.

---

The role of the values $b_j$ is as follows. When an arm $a_i$ reaches $b_k$ samples for some $k \ge 0$, it is checked for possible rejection by comparing its empirical average reward to the threshold $\tau_k$. Algorithm 3 rejects arm $i$ and moves to arm $a_{i+1}$ if the empirical average $\hat{p}_{i,b_k}$ of arm $a_i$ drops below a moving threshold $\tau_k$. The threshold $\tau_k$ begins close to $\alpha$ and gradually decreases until reaching $\beta + \varrho$ by the time $\tau_k \ge \Omega(N)$.

So for, our informal description of Alg. 3 also applies to the algorithm proposed in [GM20]. We now highlight two important differences. The first is that our algorithm is defined more carefully during the "early" phases when an arm has been sampled at most $N^{O(\varrho)}$ times. This is crucial for carrying out a rigorous analysis. The second difference is that in the main phase, we increase the sample size for a given arm in powers of $1 + \varrho$ rather than powers of 2, and also move the rejection thresholds $\tau_k$ based on the Fisher information distance via the function $\theta$. The latter ingredients allow us to obtain the optimal constant factor.

We begin the analysis of Alg 3 by proving Lemma 3.

*Proof of Lemma 3.* Let $M_j = \prod_{1 \le i \le j} Y_i$ and observe that $M_j^c$ is a positive supermartingale with $M_0 = 0$. The result follows by Doob's maximal inequality. $\qquad\square$

We will apply Lemma 3 in the following way. Let $X_i$ be the number of samples used by arm $a_i$ before rejection, and $I_i \in \{0, 1\}$ be the indicator of the event that $a_i$ is ever rejected, even if Algorithm 3 were to continue past time $N$ and sample arm $i$ an infinite number of times. We set

$$Y_i = e^{X_i} \cdot I_i,$$

With $M$ defined from $(Y_i)_{i \geq 1}$ as in Lemma 3, it follows that $\log(M)$ is at most the amount of time spent on eventual rejections before the first eventually accepted arm. Therefore if $\log(M) \leq N(1 - \varrho)$, we conclude that the last arm to be studied was sampled at least $N\varrho$ times. Since it was not rejected during that time, we can conclude this arm has $p_i \geq \beta$ with probability $1 - e^{-\Omega_\varrho(N)}$. The main contribution to the failure probability of Algorithm 3 comes from the event $\{M \geq A\}$ above, for suitable $A$. Correspondingly, the main work will be to verify $\mathbb{E}[Y_i^c] \leq 1$ for suitable $c$.

Note that $Y_i \in \{0\} \cup [1, \infty)$ almost surely for each $i$. Therefore a necessary first step in showing $\mathbb{E}[Y_i^c] \leq 1$ is to lower bound $\mathbb{P}[Y_i = 0]$, the probability that Algorithm 3 never rejects $a_i$. We now give a sufficient lower bound from the event $p_i \geq \alpha$.

**Proposition C.1.** *Let $x_1, x_2, \ldots$ be an i.i.d. Bernoulli$(p)$ sequence for $p \geq \alpha$, and let $S_k = \sum_{i=1}^k x_i$ and set*

$$\underline{S} = \inf_{k \geq 1} S_k/k.$$

*Then $\underline{S} \geq \alpha - \varrho$ holds with probability at least $c(\alpha, \varrho) > 0$. Thus $\mathbb{E}[I_i] \leq 1 - c(\alpha, \varrho)$.*

*Proof.* Since the probability that $\underline{S} \geq \alpha - \varrho$ is increasing in $p$ it suffices to take $p = \alpha$ and show the probability is positive for any $\varrho > 0$. Assume not. Then by restarting the indexing every time $S_k \leq k(\alpha - \varrho)$ holds, we find that

$$\lim_{n \to \infty} \inf S_n/n \leq \alpha - \varrho.$$

This contradicts the strong law of large numbers, thus completing the proof of the first assertion. The second assertion follows since if $S_k/k \geq \alpha - \varrho$ for all $k$ where $x_1, \ldots$ are the rewards of arm $i$, then arm $i$ will never be rejected by Algorithm 3. $\square$

Based on Proposition C.1 above, to show

$$\mathbb{E}\left[e^{X_i \cdot \frac{c_{\alpha,\beta} - \varrho_3}{\log^2 N}} \cdot I_i\right] \leq 1$$

(which is essentially what we want in light of Lemma 3), it suffices to show that

$$\mathbb{E}\left[\left(e^{X_i \cdot \frac{c_{\alpha,\beta} - \varrho_3}{\log^2 N}} - 1\right) \cdot I_i\right] \leq c(\alpha, \varrho). \tag{C.5}$$

We let $I_i^t = I_i \cdot 1_{X_i = t}$ be the event that arm $i$ was rejected after exactly $t$ steps. Since Alg 3 can only reject after $b_j$ samples, we have

$$I_i = \sum_{j=0}^{\infty} I_i^{b_j}$$

We use this to break the left-hand side of (C.5) into three separate parts and estimate the parts separately. The parts correspond to $b_0$, $b_1$ through $b_{k_0}$, and $b_{k_0 + 1}$ onward. The first two parts are easier and handled in Subsection C.3 below. The final term is the main contribution and is handled in Subsection C.4.

## C.3  Analysis of Algorithm 3 in the Small and Medium Sample Phases

Proposition C.2 bounds the contribution to (C.5) from the *small sample phase*, i.e. the first rejection condition in line 7 of Alg 3.

**Proposition C.2.** *For any $\alpha, \varrho$ there is $\varrho_1 > 0$ sufficiently small that with $b_0$ as defined above, and with $N$ sufficiently large,*

$$\mathbb{E}\left[\left(e^{X_i \cdot \frac{c_{\alpha,\beta} - \varrho_3}{\log^2 N}} - 1\right) \cdot I_i^{b_0}\right] \leq c(\alpha, \varrho)/4$$

*Proof.* It suffices to observe that for fixed $\alpha, \varrho$ and $\varrho_1$ small and $N$ sufficiently large, we have

$$e^{b_0 \cdot \frac{c_{\alpha,\beta} - \varrho_3}{\log^2 N}} - 1 \le e^{\varrho_1} - 1 \le 2\varrho_1.$$

$\square$

Proposition C.3 bounds the contribution to (C.5) from the *medium sample phase*, i.e. the second rejection condition in line 12 of Alg 3.

**Proposition C.3.** *For any $\alpha, \varrho, \varrho_1$ and for $N$ sufficiently large,*

$$\sum_{k=1}^{k_0} \mathbb{E}\left[\left(e^{X_i \cdot \frac{c_{\alpha,\beta} - \varrho_3}{\log^2 N}} - 1\right) \cdot I_i^{b_k}\right] \le c(\alpha, \varrho)/4$$

*Proof.* The event $I_i^{b_k}$ requires $|\hat{p}_{i,b_k} - \hat{p}_{i,b_{k-1}}| \ge \frac{1}{\sqrt{\log N}}$. Hence by a standard Chernoff estimate, regardless of the true reward probability $p_i$,

$$\mathbb{E}[I_i^{b_k}] \le e^{-\Omega_{\alpha,\varrho,\varrho_1}(b_k/\log N)}.$$

Since by construction $b_0 \ge \varrho_1 \log^2 N$, we have

$$\mathbb{E}\left[\left(e^{X_i \cdot \frac{c_{\alpha,\beta} - \varrho_3}{\log^2 N}} - 1\right) \cdot I_i^{b_k}\right] \le e^{b_k \frac{c_{\alpha,\beta} - \varrho_3}{\log^2 N} - \Omega_{\alpha,\varrho,\varrho_1}(b_k/\log N)}$$

$$\le e^{-\Omega_{\alpha,\varrho,\varrho_1}(\log N)}$$

$$= N^{-\Omega_{\alpha,\varrho,\varrho_1}(1)}.$$

Since $k_0 \le O(\log N)$, summing gives the desired conclusion. $\square$

Propositions C.2 and C.3 imply that the total contribution from rejections in the small and medium sample phases is at most $c(\alpha, \varrho)/2$. It remains to analyze the large sample phase in the following subsection.

### C.4 Analysis of Algorithm 3 in the Large Sample Phase

Similarly to the previous section, the main part of the analysis concerns the large sample phases $b_{k_0+j}$ for $j \ge 1$. Our goal is to precisely estimate the rejection probability at each time $b_{k_0+j}$. Note that these estimates should not depend on the true average rewards $p_i$.

Our approach is based on exchangeability and avoids any consideration of $p_i$. For a given value $j$ and a large constant $L = L(\varrho)$, consider the sequence of times

$$b_{k_0+j-L}, \; b_{k_0+j-L+1}, \; \ldots, \; b_{k_0+j}$$

and the associated sequence of empirical average rewards

$$\hat{p}_{i,b_{k_0+j-L}}, \; \hat{p}_{i,b_{k_0+j-L+1}}, \; \ldots, \; \hat{p}_{i,b_{k_0+j}}. \tag{C.6}$$

It follows from the algorithm description that for $I_i^{b_{k_0+j}}$ to occur, we must have

$$\hat{p}_{i,b_{k_0+j}} - \hat{p}_{i,b_{k_0+j-\ell}} \ge \ell \cdot \frac{d_F(\alpha,\beta)\varrho(1-\varrho_2)}{\log N}, \quad \forall\, 1 \le \ell \le L. \tag{C.7}$$

This is clear for $j > L$, but it holds also for $0 \le j \le L$ as for $N$ sufficiently large,

$$\alpha - \varrho - \frac{k_0}{\sqrt{\log N}} - L \cdot \frac{d_F(\alpha,\beta)\varrho(1-\varrho_2)}{\log N} \ge \alpha - 2\varrho.$$

By exchangeability, conditioned on the future values $\hat{p}_{i,b_{k_0+j}}, \ldots, \hat{p}_{i,b_{k_0+j-\ell}}$ the law of $\hat{p}_{i,b_{k_0+j-\ell-1}}$ depends only on $\hat{p}_{i,b_{k_0+j-\ell}}$ and is given explicitly by a hypergeometric variable. Recalling that

$R_{i,t} = n_{i,t}\hat{p}_{i,t}$ is the total reward from the first $n_{i,t}$ samples of arm $i$, $R_{i,b_{k_0+j-\ell-1}}$ has hypergeometric conditional law given by:

$$\mathbb{P}\Big[R_{i,b_{k_0+j-\ell-1}} = k \mid \big(\hat{p}_{i,b_{k_0+j}}, \ldots, \hat{p}_{i,b_{k_0+j-\ell}}\big)\Big] = \mathbb{P}\big[R_{i,b_{k_0+j-\ell-1}} = k \mid \hat{p}_{i,b_{k_0+j-\ell}}\big]$$

$$= \frac{\binom{b_{k_0+j-\ell-1}}{k}\binom{b_{k_0+j-\ell}-b_{k_0+j-\ell-1}}{R_{k_0+j-\ell}-k}}{\binom{b_{k_0+j-\ell}}{R_{k_0+j-\ell}}}. \qquad \text{(C.8)}$$

We will refer to this as the $\mathrm{HyperGeom}\big(b_{k_0+j-\ell}, b_{k_0+j-\ell-1}, R_{k_0+j-\ell}\big)$ distribution. Importantly, this distribution is independent of $\mu$. We exploit this below to control the probability of a given sequence $\big(\hat{p}_{i,b_{k_0+j-L}}, \hat{p}_{i,b_{k_0+j-L+1}}, \ldots, \hat{p}_{i,b_{k_0+j}}\big)$ of empirical average rewards. The following useful result states that hypergeometric variables automatically inherit tail bounds from the corresponding binomial random variables.

**Lemma 1** ([LP14, Hoe94]). *Fix non-negative integers $A \geq B, C$ and let $X \sim \mathrm{HyperGeom}(A, B, C)$ and $Y \sim \mathrm{Bin}(B, C/A)$. Then for any convex function $f : \mathbb{R} \to \mathbb{R}$,*

$$\mathbb{E}[f(X)] \leq \mathbb{E}[f(Y)].$$

**Lemma 2.** *For any $0 < \underline{q} < \overline{q} < 1$ and constants $\varrho > 0$ there exists $\Delta_0(\underline{q}, \overline{q}, \varrho)$ and $N_0(\underline{q}, \overline{q}, \varrho)$ such that the following holds for all $p \in [\underline{q}, \overline{q}]$. For $n \geq n_0$ sufficiently large and $\frac{1}{\Delta_0\sqrt{n}} \leq \Delta \leq \Delta_0$,*

$$\mathbb{P}\left[\frac{\mathrm{HyperGeom}(n(1+\varrho), n, np(1+\varrho))}{n} \leq p - \Delta\right] \leq e^{\left(-\frac{\Delta^2}{2p(1-p)}+\varrho\right)n}.$$

*Proof.* The corresponding binomial result Lemma 4 is proved in Theorem 2.2 in [DA92] by upper bounding an exponential moment. The same proof applies here by Lemma 1. $\qquad \square$

It will be convenient to define a restricted set of *good* sequences $(q_L, q_{L-1}, \ldots, q_0)$. These satisfy the key properties of empirical average reward sequences (C.6) for which $I_i^{b_{k_0+j}}$ holds. We say such a length $L + 1$ sequence is good if the following conditions are satisfied:

1. $q_0 \in [\underline{q}, \overline{q}] \subseteq (0, 1)$ for constants $0 < \underline{q} < \overline{q} < 1$ depending only on $\varrho, L$.
2. 
$$\max_{\ell_1, \ell_2} |q_{\ell_1} - q_{\ell_2}| \leq O\big(1/\sqrt{\log N}\big). \qquad \text{(C.9)}$$

3. For each $1 \leq \ell \leq L$:

$$\theta(q_0) \leq \theta(\alpha - 2\varrho) - j \cdot \frac{d_F(\alpha, \beta)\varrho(1 - \varrho_2)}{\log N}$$
$$\leq \theta(\alpha - 2\varrho) - (j - \ell) \cdot \frac{d_F(\alpha, \beta)\varrho(1 - \varrho_2)}{\log N}$$
$$\leq \theta(q_\ell).$$

The third condition above is necessary for $I_i^{b_{k_0+j},i} = 1$, and these together imply the first condition. Indeed for fixed $\underline{q}, \overline{q}$ and small $\varrho \in (0, 1/10)$ one always has

$$\frac{\hat{p}_{i,b_{k_0+j-1}}}{\hat{p}_{i,b_{k_0+j}}}, \frac{1 - \hat{p}_{i,b_{k_0+j-1}}}{1 - \hat{p}_{i,b_{k_0+j}}} \in \big[1 - 2\varrho, (1 - 2\varrho)^{-1}\big]$$

for large enough $N$ and any $j$. Hence it suffices to take $\underline{q} = \beta(1-2\varrho)^L$ and $\overline{q} = 1 - (1-\alpha)(1-2\varrho)^L$. With this choice, if

$$\hat{p}_{i,b_{k_0+j-L}}, \hat{p}_{i,b_{k_0+j-L+1}}, \ldots, \hat{p}_{i,b_{k_0+j}}.$$

is **not** good and $I_i^{b_{k_0+j}} = 1$, then the second condition must be the only violated one. The following easy lemma controls the failure probability of the second condition. Recall from (C.8) that conditioning on $\hat{p}_{i,b_{k_0+j}}$ determines the joint conditional law of the previous conditional rewards, regardless of $\mu$.

**Lemma 3.** *All sequences violating only the second condition* ([C.9](#)) *above have probability at most*

$$e^{-\Omega_{L,\varrho}(b_{k_0+j}/\log N)},$$

*even after conditioning on an arbitrary value for $\hat{p}_{i,b_{k_0+j}}$.*

*Proof.* The claim follows by an elementary Chernoff estimate for hypergeometric variables, which hold just as for binomial variables by Lemma [1](#). Indeed the assumption implies that some adjacent difference $|\hat{p}_{i,b_{k_0+j-\ell}} - \hat{p}_{i,b_{k_0+j-\ell+1}}|$ has size $\Omega(1/\sqrt{\log N})$. (Note for applying the Chernoff bound that $L$ is a constant independent of $N$, and so $b_{k_0+j-L} \geq \Omega_{L,\varrho}(b_{k_0+j})$.) $\square$

We now focus on upper-bounding the probability of any good sequence $(q_L, \ldots, q_0)$ appearing, conditionally on $q_0$.

**Lemma 4.** *For any good sequence $(q_L, q_{L-1}, \ldots, q_0)$ and $j \geq 0$,*

$$\mathbb{P}\Big[\big(\hat{p}_{i,b_{k_0+j-L}},\, \hat{p}_{i,b_{k_0+j-L+1}},\, \ldots,\, \hat{p}_{i,b_{k_0+j}}\big) = \big(q_L, q_{L-1}, \ldots, q_0\big) \,\big|\, p_{i,b_{k_0+j}} = q_0\Big]$$

$$\leq \exp\left(-\frac{(1 - O(\varrho))}{2q_0(1-q_0)\varrho} \sum_{\ell=0}^{L-1} b_{k_0+j-\ell}(q_\ell - q_{\ell+1})^2\right).$$

*Proof.* It suffices to show that

$$\mathbb{P}[\hat{p}_{i,b_{k_0+j-\ell-1}} = q_{\ell+1} \mid q_\ell] \leq \exp\left(-\frac{(1 - O(\varrho))}{2q_0(1-q_0)\varrho} b_{k_0+j-\ell}(q_\ell - q_{\ell+1})^2\right)$$

This follows by applying Lemma [2](#) to the hypergeometric random variable

$$\hat{p}_{i,b_{k_0+j-\ell}} \cdot b_{k_0+j-\ell} - \hat{p}_{i,b_{k_0+j-\ell-1}} \cdot b_{k_0+j-\ell-1} = R_{b_{k_0+j-\ell}} - R_{b_{k_0+j-\ell-1}}.$$

The fact that

$$b_{k_0+j-\ell+1} - b_{k_0+j-\ell} = \varrho \cdot b_{k_0+j-\ell} \pm O(1)$$

leads to the factor of $\varrho$ in the denominator of the desired result. $\square$

**Lemma 5.** *For fixed problem parameters and $N$ large, any good sequence $(q_L, \ldots, q_0)$ satisfies*

$$q_\ell \geq q_0 + \frac{\ell \cdot d_F(\alpha, \beta)\varrho(1 - 2\varrho_2) \cdot \sqrt{q_0(1-q_0)}}{(\log N)}$$

*Proof.* Recall that $\theta'(q) = \frac{1}{\sqrt{q(1-q)}}$ and that $\theta$ is smooth on $[\underline{q}, \overline{q}] \subseteq (0, 1)$. By Item 2 above, all $q_\ell$ are within $o_N(1)$ of each other, so the result follows from the inverse function theorem. (Notice that the factor $(1 - \varrho_2)$ changed to $(1 - 2\varrho_2)$ above.) $\square$

**Lemma 6.** *For $1 \leq m \leq L$ and any good sequence $(q_L, \ldots, q_0)$, we have*

$$\sum_{\ell=0}^{m-1} (q_\ell - q_{\ell+1})^2 \geq \frac{m \cdot d_F(\alpha, \beta)^2 \varrho^2 (1 - 4\varrho_2) \cdot q_0(1-q_0)}{\log^2 N}.$$

*Proof.* The result follows from Lemma [5](#) and Cauchy-Schwarz in the form

$$\sum_{\ell=0}^{m-1} (q_\ell - q_{\ell+1})^2 \geq m^{-1} \left(\sum_{\ell=0}^{m-1} |q_\ell - q_{\ell+1}|\right)^2.$$

$\square$

**Lemma 7.** *For any good sequence $(q_L, \ldots, q_0)$ and $j \geq 0$, we have*

$$\sum_{\ell=0}^{L-1} b_{k_0+j-\ell}(q_\ell - q_{\ell+1})^2 \geq (1 - O(\varrho_2)) \cdot \frac{b_{k_0+j}\varrho \, d_F(\alpha, \beta)^2 \cdot q_0(1-q_0)}{\log^2 N}.$$

*Proof.* We break the sum into parts and apply Lemma 6 to each one. We have:

$$\sum_{\ell=0}^{L-1} b_{k_0+j-\ell}(q_\ell - q_{\ell+1})^2 = b_{k_0+j-L+1}\sum_{\ell=0}^{L-1}(q_\ell - q_{\ell+1})^2 + \sum_{m=1}^{L-1}(b_{k_0+j-m+1} - b_{k_0+j-m})\sum_{\ell=0}^{m-1}(q_\ell - q_{\ell+1})^2$$

$$\geq \sum_{m=1}^{L-1} b_{k_0+j} \cdot \frac{\varrho}{(1+\varrho)^{m+10}} \cdot (1 - 4\varrho_2)\frac{m\varrho^2 d_F(\alpha,\beta)^2 \cdot q_0(1-q_0)}{\log^2 N}$$

$$\geq (1 - O(\varrho + \varrho_2)) \cdot b_{k_0+j} \cdot \frac{\varrho^3 d_F(\alpha,\beta)^2 \cdot q_0(1-q_0)}{\log^2 N} \cdot \sum_{m=1}^{L-1}\frac{m}{(1+\varrho)^m}.$$

For $L = L(\varrho) = O(\varrho^{-1}\log(\varrho^{-1}))$ sufficiently large,

$$\sum_{m=1}^{L-1}\frac{m\varrho}{(1+\varrho)^m} \geq (1-\varrho)\sum_{m=1}^{\infty}\frac{m}{(1+\varrho)^m}.$$

$$= (1-\varrho)\left(\sum_{m=1}^{\infty}\frac{1}{(1+\varrho)^m}\right)^2$$

$$= \frac{1-\varrho}{\varrho^2}.$$

Substituting and recalling that $\varrho \ll \varrho_2$ completes the proof. $\qquad\square$

Combining with Lemma 4 yields the second inequality below (the first is trivial).

**Corollary C.4.** *For any $\mu$ and $q_0$, we have*

$$\mathbb{P}^{p_i \sim \mu}\left[\left(\hat{p}_{i,b_{k_0+j-L}},\ \hat{p}_{i,b_{k_0+j-L+1}},\ \ldots,\ \hat{p}_{i,b_{k_0+j}}\right) = \left(q_L, q_{L-1}, \ldots, q_0\right)\right]$$

$$\leq \mathbb{P}\left[\left(\hat{p}_{i,b_{k_0+j-L}},\ \hat{p}_{i,b_{k_0+j-L+1}},\ \ldots,\ \hat{p}_{i,b_{k_0+j}}\right) = \left(q_L, q_{L-1}, \ldots, q_0\right) \mid p_{i,b_{k_0+j}} = q_0\right]$$

$$\leq \exp\left(-\left(1 - O(\varrho_2)\right)\frac{b_{k_0+j}d_F(\alpha,\beta)^2}{2\log^2 N}\right).$$

**Lemma 8.** *Let $j_0$ be the largest $j$ such that $b_{k_0+j} \leq N$. Then for $N$ sufficiently large,*

$$\sum_{j=1}^{j_0}\mathbb{E}\left[e^{X_i \cdot \frac{c_{\alpha,\beta}-\varrho_3}{\log^2 N}} \cdot I_i^{b_{k_0+j}}\right] \leq c(\alpha,\varrho)/4.$$

*Proof.* Recall that $c_{\alpha,\beta} = \frac{d_F(\alpha,\beta)^2}{2}$, and observe that the number of total sequences $(q_L, \ldots, q_0) \in [0,1]^{L+1}$ with $b_{k_0+j+\ell}q_\ell \in \mathbb{Z}$ is at most $N^{L+1}$ for each $j \leq j_0$. Combining Lemma 3 and Corollary C.4 and noting that the latter always gives the main contribution, we find for each $j \leq j_0$,

$$\mathbb{E}\left[e^{X_i \cdot \frac{c_{\alpha,\beta}-\varrho_3}{\log^2 N}} \cdot I_i^{b_{k_0+j}}\right] \leq N^{L+1}\exp\left(\frac{b_{k_0+j}}{\log^2 N} \cdot \left((c_{\alpha,\beta} - \varrho_3) - (1 - O(\varrho_2))c_{\alpha,\beta}\right)\right)$$

$$\leq \exp\left(-\Omega\left(\frac{\varrho_3 b_{k_0+j}}{\log^2 N}\right)\right)$$

so long as $\varrho_3$ is chosen so that $\varrho_3 \gg \max(\varrho, \varrho_2)$. In the last line we used the fact that $b_{k_0+j} \geq b_{k_0} \geq \log^4 N$ to absorb the factor $N^{L+1} \leq e^{\varrho \log^{3/2} N}$ for large $N$. Summing over $j$ gives the

desired result, since for $\varrho_4 = \Omega(\varrho_3)$ and $N$ sufficiently large,

$$
\begin{aligned}
\sum_{j=1}^{\infty} e^{-\Omega\left(\frac{\varrho_3 b_{k_0+j}}{\log^2 N}\right)} &\leq \sum_{m=1}^{\infty} e^{-\frac{\varrho_4(m+b_{k_0})}{\log^2 N}} \\
&= e^{-\varrho_4 \log^2 N} \sum_{m=1}^{\infty} e^{-\frac{\varrho_4 m}{\log^2 N}} \\
&\leq e^{-\varrho_4 \log^2 N} \cdot O\left(\frac{\log^2 N}{\varrho_4}\right) \\
&\leq e^{-\frac{\varrho_4 \log^2 N}{2}} \\
&\leq c(\alpha, \varrho)/4.
\end{aligned}
$$

$\square$

We now use Lemma 3 to conclude.

*Proof that Algorithm 3 achieves the guarantee of Theorem 3.1.* By combining Lemma 8 with the previous Propositions C.2 and C.3, it follows that

$$
\mathbb{E}\left[e^{X_i \cdot \frac{c_{\alpha,\beta} - \varrho_3}{\log^2 N}} \cdot I_i\right] \leq 1.
$$

Lemma 3 now implies that the total amount of time spent on eventually rejected arms is at most $N(1-\varrho)$ with probability

$$
e^{-\frac{(c_{\alpha,\beta} - \varrho_3)(1-\varrho)N}{\log^2 N}}.
$$

On this event, the output arm $i^*$ satisfies $n_{i^*,N} \geq \varrho N$ by definition. Since $i^*$ was not rejected, for $j_1$ be the largest value such $b_{k_0+j_1} \leq \varrho N$ we have

$$
\hat{p}_{i^*, b_{k_0+j_1}} \geq \beta + \varrho.
$$

The probability for this to hold if $p_i \leq \beta$ is at most $e^{-\Omega_\varrho(N)}$. Altogether we find that

$$
\mathbb{P}[p_{i^*} \geq \beta] \geq 1 - \exp\left(-\frac{(c_{\alpha,\beta} - \varrho_5)N}{\log^2 N}\right) - e^{-\Omega_\varrho(N)} \tag{C.10}
$$

for $\varrho_5$ arbitrarily small. This concludes the analysis of Algorithm 3 (since the last error term is negligible). $\square$

## C.5 Finding Many Good Arms with a Fixed Budget

In this final subsection we observe that Algorithm 3 can be modified to output as many as $\log N$ distinct arms each of which satisfies the same $(\eta, \varepsilon, \delta)$-PAC guarantee[2], with no degradation in the asymptotic failure probability. With other parameters fixed, we denote the $N$-sample version of Algorithm 3 by $\mathcal{A}_N$ to emphasize the dependence on $N$. In particular, $N$ both equals the number of steps in $\mathcal{A}_N$ and appears (via its logarithm) in the description of $\mathcal{A}_N$'s individual steps.

Let $\tilde{N} = N + \lceil \frac{2N}{\log^{1/2}(N)} \rceil$. We consider a modified algorithm $\tilde{\mathcal{A}}_{\tilde{N}}$ which mimics the behavior of $\mathcal{A}_N$ with two changes:

1. $\tilde{\mathcal{A}}_{\tilde{N}}$ is a $\tilde{N}$-sample algorithm.

2. If an arm $a_i$ has not yet been rejected after $M = \lceil N/\log^{3/2}(N) \rceil$ samples, then $\tilde{\mathcal{A}}_{\tilde{N}}$ accepts $a_i$ and continues to $a_{i+1}$. In particular, $\tilde{\mathcal{A}}_{\tilde{N}}$ may accept several arms instead of just one.

**Theorem C.9.** *With probability* $1 - \exp\left(-\frac{(c_{\alpha,\beta} - \varrho_5 - o_N(1))N}{\log^2 N}\right)$, $\tilde{\mathcal{A}}_{\tilde{N}}$ *accepts at least* $\log(N)$ *distinct arms* $a_i$, *all of which satisfy* $p_i \geq \beta$.

---

[2]In fact $\log N$ can be replaced by anything $o_N(\log^2 N)$ by more precisely defining $M$ and $\tilde{N}$.

The change from $N$ to $\tilde{N}$ is almost irrelevant in the actual statement of Theorem C.9 since $\log(N) \geq \log(\tilde{N}) - o_N(1)$. In particular, $\tilde{\mathcal{A}}_{\tilde{N}}$ is a $\tilde{N}$-sample algorithm which outputs at least $\log(\tilde{N}) - 1$ arms with probability $1 - \exp\left(-\frac{(c_{\alpha,\beta} - \varrho_5 - o_{\tilde{N}}(1))\tilde{N}}{\log^2 \tilde{N}}\right)$. It is certainly not really necessary to use the value $\log(N)$ rather than $\log(\tilde{N})$ to describe the individual steps taken by $\tilde{A}_{\tilde{N}}$. However introducing $\tilde{N}$ streamlines the proof below by letting us treat $\mathcal{A}_N$ as a blackbox.

*Proof.* To show that all accepted arms $a_i$ satisfy $p_i \geq \beta$ with sufficiently high probability, it suffices to consider (C.10) with the final term replaced by $e^{-\Omega_\varrho(N/\log^{3/2}(N))}$. In particular, observe that the main term does not change, even after multiplying the failure probability by $O\left(\log^{3/2}(N)\right)$ (the maximum possible number of arms accepted by $\tilde{\mathcal{A}}_{\tilde{N}}$. Thus we focus on showing that $\tilde{\mathcal{A}}_{\tilde{N}}$ outputs at least $\log(N)$ arms with high probability.

Consider yet another $N$-sample algorithm $\widehat{\mathcal{A}}_N$ which deletes each arm independently with probability $1/N$ and follows $\mathcal{A}_N$ on the set of non-deleted arms in order of increasing index. (Like $\mathcal{A}_N$, $\widehat{\mathcal{A}}_N$ never accepts arms before time $N$.) We simulate $\tilde{\mathcal{A}}_{\tilde{N}}$ and $\widehat{\mathcal{A}}_N$ on the same reward sequences, i.e. we couple them so that the $t$-th sample of arm $a_i$ always gives the same result for each $(t,i)$. We **claim** that in this coupling, conditioned on $\tilde{\mathcal{A}}_{\tilde{N}}$ failing to accept $\log(N)$ arms within the first $\tilde{N}$ samples, $\widehat{\mathcal{A}}_N$ has probability $\Omega(N^{-\log(N)})$ to fail (i.e. output $a_i$ with $p_i < \beta$) when run for $N$ samples.

First let us assume the claim and deduce Theorem C.9. Denote by $p(N)$ the probability for $\mathcal{A}_N$ to fail. Note that $\widehat{\mathcal{A}}_N$ has the same failure probability $p(N)$, having in fact the same behavior as $\mathcal{A}_N$ in distribution (as the set of deleted arms is independent of everything else). Moreover let $\tilde{p}(\tilde{N}, k)$ denote the probability that $\tilde{\mathcal{A}}_{\tilde{N}}$ fails to accept at least $k$ arms. The claim above implies that

$$\begin{aligned} \tilde{p}(\tilde{N}, \log N) &\leq O\left(N^{\log N}\right) \cdot p(N, 1) \\ &\leq e^{o_N(N/\log^2 N)} \cdot p(N, 1) \\ &\leq \exp\left(-\frac{(c_{\alpha,\beta} - \varrho_5 - o_N(1))N}{\log^2 N}\right). \end{aligned}$$

It remains to prove the above claim. Let us say the infinite i.i.d. reward sequence $(r_{i,n})_{n \geq 1}$ of arm $a_i$ is **acceptable** if $\mathcal{A}_N$ would not reject $a_i$ within $M$ samples, i.e. $\tilde{\mathcal{A}}_{\tilde{N}}$ will either accept $a_i$ or run out of samples before doing so. We take the point of view that each $a_i$ is either acceptable or not (by randomly fixing the reward sequences at the start). Then with probability $\Omega(N^{-\log(N)})$, the first $\log(N)$ acceptable arms are skipped by $\widehat{\mathcal{A}}$, and the first $\hat{N}$ unacceptable arms are not skipped. On this event, the first $\hat{N} - M \geq N$ samples obtained by $\widehat{\mathcal{A}}_N$, i.e. all $N$ of its samples, are drawn from unacceptable arms. On this event, $\widehat{\mathcal{A}}_N$ fails with constant probability, which establishes the claim and completes the proof. $\square$

