# OpenReview forum: "Asymptotically Optimal Quantile Pure Exploration for Infinite-Armed Bandits"
_NeurIPS.cc/2023/Conference — NeurIPS 2023 poster_

### Official Review · Reviewer_ccFe · 2023-06-21

**Soundness:** 2 fair
**Presentation:** 2 fair
**Contribution:** 3 good
**Rating:** 7
**Confidence:** 3

**Summary:**

This paper addresses a problem of pure exploration in the infinite-armed bandit setting : in this problem, each arm has mean reward $p_i$ drawn i.i.d. from a distribution with quantile function $G^{-1}$, and the aim is to output an arm $a_{i*}$ after sampling $N$ times an action, such that $\mathbb{P}\left(p_{i^*} \geq G^{-1}(1 - \eta) - \epsilon\right)\geq 1 - \delta$ for some given $\eta$, $\epsilon$, and $\delta$. The authors first propose an algorithm for the setting where $G^{-1}(1 - \eta)$ is unknown that is almost optimal in terms of expected sample complexity $\mathbb{E}[N]$, and show that is satisfies the $(\eta, \epsilon, \delta)$-PAC constraint and derive an upper bound on the sample complexity. In a second part of the article, they consider the fixed budget setting when $G^{-1}(1 - \eta)$ is unknown, and propose an asymptotically optimal algorithm (i.e. an algorithm with asymptotically optimal failure rate). They derive upper bound on its failure rate, and obtain a lower bound for the failure rate of any algorithm.

**Strengths:**

Originality : The author consider in this paper a vary natural problem.

Clarity : This paper is very well written and easy to follow. The literature review is thorough and well presented, the algorithms are rather well explained. The comments on the results are interesting, and the authors manage to highlight the different aspects of the problem.

Quality : This paper provides interesting theoretical results.

Significance : This article makes significant contributions to a natural problem. They improve on existing results, and close open questions in both settings by characterizing optimal sample complexity and optimal failure rate in the fixed confidence and fixed budget setting.

**Weaknesses:**

Results are sometimes lacking in clarity, and sometimes even in rigor. Bellow are some examples :

The algorithms make use of some undefined constant $C$ (the same holds for the constants $\rho, \rho_1, \rho_2$). While I agree that the results are true for some specific constant, they are not true in general. The authors should define these constants more carefully, and propagate the numerical constants appearing in their proof instead of using symbols $\Omega$. The authors make heavy use of these symbols  in their proof, making them harder to read, and less rigourous.

I find the "proof" of Lemma 4 not convincing.


**Questions:**

Please define explicitly the constant to use in the algorithms so that your claims hold.

Please postpone the proof of Lemma 4 to the Appendix and develop your arguments further.

For the sake of completeness, please recall the results that you are using (Chernoff bounds (A.1) and (A.2), Thm 4.5 in [MU17], Thm 1 in [GM14], etc).

Please refrain from using symbols as $\Omega$ in the proofs, and use instead explicit numerical constants, so that the validity of your arguments is easier to assess.

There are somme remaining typos :
- Theorem 1.2: the constant in the definition of $N$ should be $c_{\alpha, \beta}$
- Algorithms 1, 2: $C$ is not defined.
- Algorithm 3: $\alpha$, $\rho$, and maybe somme other parameters should be considered as input to the algorithm
- line 167 : what is $\eta_1$? Is it $\eta$?
- Algorithm 1 : why not define $K$ as $K = C\log(1/\delta)/\eta$?
- line 199: do you mean Theorem 2.1?
- Appendix, line 474 : there is a "with" missing
- Appendix, line 490 : "qualitative"
- Appendix, Lemma 4: there are extra parentheses.
- Appendix, line 621 : please define the event $E$

I had trouble understanding the sentence in line 197 (I think the word therefore is misleading); could you please rephrase it?

The meaning of the number of "batches" is not very easy to understand. Could you please rephrase the corresponding discussions?

It would be nice to include an outline and a conclusion in this article.

Theorem 2.1 and Algorithm 1 overlap, which is annoying. Please correct the layout.

You introduce constants $\rho_1, \rho_2$ which do not appear in Algorithm 3. Please correct that.

**Limitations:**

The authors have not discussed limitations of their results.

---

> ### Author Rebuttal · Authors · 2023-08-09
>
> We thank the reviewer for their time and insightful reviews. Here is our detailed response.
>
> Thank you for all of the helpful suggestions. We have taken them carefully into account while revising the paper.
>
> Typos:
>
> 1. Thank you! We have corrected the typo.
>
> 2. We apologize. C is a universal constant (basically the same as writing O(), but using C emphasizes that there is no dependence on any parameters at all). We have added some explanation in the early parts of the paper.
>
> 3. Thank you for the suggestion! We have added the parameter input line in Alg 3.
>
> 4. Thank you! We have corrected the typo.
>
> 5. We have adopted your suggestion. Thank you!
>
> 6. Thank you! We have corrected the typo.
>
> 7. Thank you! We have added the “with”.
>
> 8. Thank you! We have corrected the typo.
>
> 9. The extra parentheses were supposed to show dependence on other parameters similarly to the rest of Lemma 4. We fixed it, thank you for catching this typo.
>
> 10. Thank you for the suggestion. We have written out the definition of the event E. We also changed $\delta$ to $\Delta_N$ in Equation (B.14) just above.
>
> Other comments:
> 1. Thank you for your comment. We have rewritten the proof of Lemma 4 and included it in the 1 page pdf. Previously our point of view in the proof was “fully Bayesian” for each fixed algorithm. The proof is short and valid once understood properly, but requires some precision about which probability spaces are being used. We didn’t specify, which probably caused the confusion, and feel it would be cumbersome to do so. We now explain it by defining a value function $E_t$ and using the dynamic programming principle instead of fixing a specific algorithm. We hope it is now clearer and would be happy to provide further explanations or changes.
>
> 2. Line 197: we updated the paragraph. The meaning is just that the upper and lower bounds $U,L$ always satisfy $U\leq O(L\log L)$, i.e. the bounds can be said to nearly match.
>
> 3. Thank you for the question! We have rephrased the remark about batches, and included it in the 1-page pdf.
>
> 4. Thanks! Theorem 2.1 and Algorithm 1 no longer overlap. We also made Algorithms 1,2 adjacent to mitigate these annoyances.
>
> 5. We added a conclusion, thank you for the suggestion.
>
> 6. $\varrho_1$ appears in the definition of $b_0$. We updated Algorithm 3 to reference these parameters at the start.

---

> > ### Comment · Reviewer_ccFe · 2023-08-14
> >
> > I thank the authors for their response, which addresses my previous concerns. I have raised my score.

---

### Official Review · Reviewer_326s · 2023-06-29

**Soundness:** 3 good
**Presentation:** 2 fair
**Contribution:** 3 good
**Rating:** 6
**Confidence:** 4

**Summary:**

This paper considers pure exploration problem in multi-armed bandits with infinite number of arms. Its goal is to return an arm with sufficiently high expected reward (within the top $\eta$ region) with high probability (at least $1-\delta$). A naive method could lead to a complexity upper bound of $O({\log(1/\eta\delta)\log(1/\delta) \over \eta\epsilon^2})$, where $\epsilon$ is the error tolerance rate. The authors make some improvement, design a new learning algorithm, and show that it achieves a complexity upper bound of $O({\log(1/\eta)\log(1/\delta) \over \eta\epsilon^2})$. They also consider the fixed budget case, and show that their proposed algorithm can achieve a failure probability approximately $\exp(-N/\log^2 N)$, where $N$ is the fixed budget.

**Strengths:**

The model setting is well-motivated, and reducing one $\log(1/\delta)$ factor seems to be an important step in this kind of problems.


**Weaknesses:**

Here are some of my questions.

1) An $(\eta, \epsilon, \delta)$-PAC is defined as Eq. (1.1), i.e., $\Pr[p_{i^*} \ge G^{-1}(1-\eta) -\epsilon] \ge 1 - \delta$. I can understand that this is a traditional definition for $\epsilon$-PAC in pure exploration problems, but I am still wondering what if we change this definition to be $\Pr[p_{i^*} \ge G^{-1}(1-\eta - \epsilon)] \ge 1 - \delta$. How would this change influence all the complexity bounds?

2) In Theorem 1.2, does the $\alpha \ge \eta$ mean $\alpha \ge G^{-1}(1-\eta) $? Is the $c$ in equation before Eq. (1.2) the same as the $c_{\alpha, \beta}$ in Eq. (1.2)?

3) I do not really understand why $\eta$ does not appear (in the statement of line 75). In my opinion, a natural lower bound should be $(1-\eta)^N$, since this is the probability that you try $N$ arms and no one satisfy the constraint that $p_i \ge G^{-1}(1-\eta)$. Is this hidden by the factor $c$ (or $c_{\alpha, \beta}$)?

4) In Algorithm 1 line 2, why we are writing an $\eta$ factor on both the numerator and the denominator?

5) In Algorithm 2, each arm should be pulled for $O({\log(1/\eta\delta) \over \epsilon^2})$ times, and try $O({\log(1/\delta) \over \eta})$ arms. This leads to a straightforward $O({\log(1/\eta\delta)\log(1/\delta) \over \eta\epsilon^2})$ complexity bound. I guess the reason that the complexity bound in Theorem 2.1 is $O({\log(1/\eta)\log(1/\delta) \over \eta\epsilon^2})$ is because the expected number of tried arms (and then find one good arm to output) is about $O({1 \over \eta})$, but this seems not leading to the complexity bound in Theorem 2.1. Since  no proof sketch about this theorem is given in the main text, I think at least you need to explain the high-level ideas for this proof clear to make sure that the readers can understand why your theorem is true. Besides, is your complexity bound only an expected one or a high-probability one?

6) In line 281, do you mean $\log M \le N(1-1/\log N)$ instead of $M \le N(1-1/\log N)$?

**Questions:**

Please see the above "Weaknesses"

---

> ### Author Rebuttal · Authors · 2023-08-09
>
> We thank the reviewer for their time and insightful reviews. Here is our detailed response.
>
> We would like to emphasize that we not only give an algorithm achieving failure probability $\exp(-N/\log^2 N)$, but we also show this is unimprovable in a natural asymptotic parameter regime. This lower bound is what we consider the most surprising finding of our work and seems to be very unusual; one typically expects the sample complexity to be proportional to $\log(1/\delta)$. We are not aware of another learning problem exhibiting the rate $\log(1/\delta)\big(\log\log 1/\delta\big)^2$ we found.
>
> 1. This is a good question, and we have added clarification slightly above Theorem 1.1. In short, the modified definition also leads to an unsolvable problem. Imagine 90% of arms have mean 0.5, while 10% of arms have mean $0.5-e^{-N^4}$. Then $G^{-1}(1-\eta-\epsilon) = 0.5$ so long as $\eta+\epsilon < 0.9$. So if there is only quantile slack, one needs to distinguish between the two means to solve the problem, which is information theoretically impossible.
>
> 2. Thank you for pointing these out. We apologize for the typos and have corrected them. We meant to write $\alpha ≤ G^{-1}(1-\eta)$. The point is that we are given a target mean reward $\alpha$, and are promised that at least $\eta$ fraction of arms have mean reward $\alpha$ or better.  The c should also be $c_{\alpha,\beta}$ as you point out.
> The reason for $\eta$ to be irrelevant for the fixed budget setting is that we are studying the limit of small $\delta$, i.e. very high confidence. For any fixed $\eta$, the bound you suggest is indeed valid and shows the failure probability is at least $\exp(-\eta N)$. (Corresponding to sample complexity at least $\log(1/\delta)/\eta)$.
>
> 3. The surprising thing is that the correct failure probability is of larger order in N, i.e. the required sample complexity is of larger order in $\delta$. So for any fixed $(\alpha,\beta,\eta)$, as $\delta\to 0$ the relevant constant factor $c_{\alpha,\beta}$ does not depend on $\eta$. We added some further explanation of this point just above Theorem 3.2 and included some intuition in the 1-page pdf response. It is certainly interesting to get more precise bounds that allow $\eta,\varepsilon,\delta$ to tend to zero simultaneously; in such cases, our lower bound $\log(1/\delta)(\log\log 1/\delta)^2$ would always be valid.
>
> 4. Thank you! We have corrected the typo.
>
> 5. Thank you for the great suggestion. We have added some explanation in the paper. You are exactly correct that the expected number of arms tried in Algorithm 2 is $O(1/\eta)$. The sample complexity actually ends up being dominated by Algorithm 1, which collects $\log(1/\eta)$ samples each from $\log(1/\delta)$ arms.
>
> 6. Thank you! We have corrected the typo.

---

> > ### Comment · Reviewer_326s · 2023-08-14
> > **Thank you**
> >
> > Thanks for your explanation. I will increase my score to 6.

---

### Official Review · Reviewer_o2AR · 2023-07-06

**Soundness:** 4 excellent
**Presentation:** 3 good
**Contribution:** 4 excellent
**Rating:** 8
**Confidence:** 2

**Summary:**

This paper studies pure exploration problem in bandits with infinite arms, where the mean reward of each arm is i.i.d. according to some unknown distribution. In fixed-confidence setting, this paper proposes an algorithm that nearly matches the optimal sample complexity. In fixed-budget setting, this paper proposes an asymptotic lower bound and proposes an algorithm that nearly matches it.

**Strengths:**

- The fixed-confidence algorithm is nearly optimal.
- The lower bound proof for fixed-budget setting contains novel construction of adversaries.
- The fixed-budget algorithm is asymptotically optimal and the limitation on the knowledge of $\alpha$ is appropriately addressed.

**Weaknesses:**

Is that possible to empirically evaluate the proposed algorithms, especially the fixed-budget algorithm, and compare them with previous work?

### Suggestions on Writing
- line 213, "known" -> "unknown".
- It may be better to mention the low probability of faliure when discussing Algorithm 2.
- It may be better to replace $\mathbb{P}$ in equations (3.1) and (3.2) by $\mathbb{P}_{\mu_N}$.
- It seems $\hat{p}_{i, b_k}$ has a slight abuse of notation compared to the definition in Section 1.2 since $b_k$ in Algorithm 3 is not the time index from $t=1$. Maybe some clarification can be added here.
- What does $\partial v$ mean in the definition 3.2??

**Questions:**

- Is that possible to build some connection with the instance-dependent fixed-budget complexity in pure exploration with finite arms?
- Since $\eta$ does not appear in the error probability of fixed-budget setting, can we just take $\eta=0$?
- Is that possible to intuitively explain why Algorithm 3 will be optimal?
- Why does the ceiling function appear in $b_{k_0+j}$ but not in $b_k$?
- Why will an adversary with smaller strength make the true failure probability of $\mathcal{A}$ larger?

**Limitations:**

The limitations (in particular, the requirement of the knowledge of $\alpha$) has been adequately addressed.

---

> ### Author Rebuttal · Authors · 2023-08-09
>
> We thank the reviewer for their time and insightful reviews. Here is our detailed response.
>
> Suggestions:
> 1. For this suggestion, we actually intended to write “known” because we are discussing the lower bound. This “known” makes pure exploration easier, hence the lower bound is stronger. We added a brief clarification of this in the paper.
>
> 2. Thank you for the suggestion! We have adopted your suggestion.
>
> 3. Thank you for the suggestion! We have adopted your suggestion
>
> 4. Thanks, we have updated the notation.
>
> 5. Thank you! We apologize for the typo (it was supposed to be $\mathbb A$) and have fixed it.
>
>
> Questions:
>
> 1. This is an interesting question and they certainly appear similar. We believe the behavior is somewhat different at a fine-grained level. This can be seen from https://arxiv.org/abs/2303.09468, which shows that there is no complexity measure characterizing optimal pure exploration with finitely many arms. This means that the instance-optimal sample complexities for known arm distributions cannot be simultaneously achieved (with sharp constant) for an unknown arm distribution. On the other hand, our results for the fixed budget setting obtain a sharp constant, and the algorithm is agnostic to the reservoir distribution as long as N is sufficiently large. The difference seems to stem from the fact that in an infinite arm setting, one might get unlucky and end up sampling mostly bad arms. This turns out to be a steeper barrier to success than learning the target quantile (see Appendix C.1). Meanwhile for K arms, estimating the target quantile is of the same “order” of difficulty as solving the problem so they cannot be disentangled.
>
> 2. The error probability holds for any fixed $\eta>0$, but $\delta$ must be small enough (or N large enough) depending on $\eta$. If $\eta$ were to shrink rapidly with $\delta$, then the results might be different.
>
> 3. Thank you for the question, and here is some intuition. The algorithm is inspired by the lower bound proof, where an adversary forces each arm to degrade in performance as it is sampled more. This “degradation schedule” is identical to the “rejection threshold schedule” in the algorithm. In the lower bound, there is clearer intuition: the adversary wants to use a constant rate of probability distortion “budget” per sample.
>
> 4. Thank you! We have corrected the typo.
>
> 5. The important thing is to keep in mind that an adversary is nothing more than a proof technique. A weak adversary distorts the distribution of the algorithm’s trajectory by a small amount. So if a weak adversary is able to force failure with probability 1, it is natural that this implies a large value of the true failure probability.

---

> > ### Comment · Reviewer_o2AR · 2023-08-14
> > **Response**
> >
> > Thank you very much for your rebuttal and my concerns have been addressed.

---

### Official Review · Reviewer_xhAx · 2023-07-06

**Soundness:** 3 good
**Presentation:** 3 good
**Contribution:** 4 excellent
**Rating:** 6
**Confidence:** 1

**Summary:**

**[Setting]** This paper studies best arm identification among countably infinite arms in both fixed confidence and fixed budget setting. The mean rewards for these arms are drawn iid from a distribution $\mu$. Let $\alpha$ be the $1 - \eta$ percentile of $\mu$. The goal is to find an arm with mean reward at least $\alpha - \epsilon$. Note that the algorithm knows $\eta$ and not $\alpha$.

**[Algorithm - fixed confidence]** - In the fixed confidence setting, the authors propose an algorithm that identifies the required arm with high probability using $O(\log(1/\delta) \log (1/\eta) / \eta \epsilon^2)$ samples in expectation. This algorithm first tries to estimate $\alpha$ by pulling $O(\log (1/\delta) / \eta)$ arms $O(\log(1/\eta) / \epsilon^2)$ times, and then uses the estimated value to find the target arm by pulling $O(1)$ arms (in expectation) $O(\log (1 / \eta \delta) / \epsilon^2)$ times.

**[Guarantee - fixed confidence]** - The bound matches matches a known lower bound up to $\log (1 / \eta)$ factor. This is an improvement over previous works that include a $(\log (1 / \delta))^2$ term.

**[Algorithm - fixed budget]** - The authors propose an algorithm that goes though arms in a sequence. For each arm $i$, samples are collected in batches until this arm is discarded because its mean falls ``reasonably'' below $\alpha$. The threshold for discarding an arm keeps decreasing across batches, but always stays above $\alpha - \epsilon$. This ensures that arms with mean < $\alpha - \epsilon$ are discarded early on using few samples, but arms with mean > $\alpha - \epsilon$ have to increasingly pass a harder test to be retained. The last un-discarded arm is returned as the best arm.

**[Guarantee - fixed confidence]** - The authors focus on the case where $\delta \rightarrow 0$ and derive matching lower and upper bounds showing that an optimal algorithm in the fixed budget setting needs $O(\log (1/\delta) (\log \log(1 / \delta))^2 / c)$ samples, where $c$ is a problem dependent parameter.

**Strengths:**

**Originality** - The paper closes a gap (w.r.t $\delta$) between the upper and lower bounds in the fixed confidence setting. This requires new ideas in the algorithm. It also proposes the first asymptotically optimal algorithm in the fixed budget setting.

**Quality** - I must admit that I did not fully understand the results in the fixed budget setting. For the fixed confidence setting, the ideas appear technically sound to me, though I have a few questions listed later. This is a complete study in my opinion where the derived bounds are tight, and the claims are well supported with theoretical evidence.

**Clarity** - The paper is positioned well with respect to other works. Most parts are very well written and organised (barring a few basic questions listed later).

**Significance** - Given that the paper closes the gap between upper and lower bounds in both fixed confidence and fixed budget setting, it would be a valuable addition to the literature. It also provides a good starting point for further explorations like getting high probability bounds on the sample complexity in the fixed confidence setting.


**Weaknesses:**

**Quality** -

1. As the authors have pointed out in L212-215, the derived bounds do not depend on $\eta$ in the fixed confidence setting. However, the intuition for why this is the case is missing from the paper.

2. L206 says that the case of known $\alpha$ is the easier case. Remark 3.1 gives conditions under which one would not need $\alpha$ in advance. However, it appears that Algorithm 3 always requires $\alpha$ as input (see L6 for example). This must be clarified.


**Clarity** - The paper can be made more accessible in many places:

1. It is perhaps just me, but the sample complexity bounds for the fixed budget setting are a bit confusing. Isn't it true that people usually fix a budget $N$ and try to minimise the probability of error with this budget? What does the case $\delta \rightarrow 0$ mean then? Why is there a bound on $N$ again (which is supposed to be fixed)?

2. The idea of batches in Remark 2.1 can be explained more clearly. In the worst case, Algorithm 2 does sample $O(\log(1/\delta))$ arms. Using batches probably ensures that only $O(1)$ arms are queried in expectation.

3. The notation is confusing at several places. For example, $\eta_1$ in L167 is undefined. L2 in Algorithm 1 uses $\eta$ in both numerator and denominator. $\eta_2$ in L184 should probably be explicitly defined.

4. Should there be an $\eta$ in the denominator inside the $\log$ in L185? This will make $b_{2, i}
 = C \log(1/\delta) / \eta^2$, which is different from the range of the for loop in Algorithm 2.


**Questions:**

Please address points 1 and 2 under quality and points 1 and 4 under clarity in the weakness section.

**Limitations:**

I believe that point 2 under clarity in the weaknesses section should be addressed.

---

> ### Author Rebuttal · Authors · 2023-08-09
>
> We thank the reviewer for their time and insightful reviews. Here is our detailed response.
>
> Quality Questions:
> 1. L212-215 are comments for the fixed budget setting (as opposed to the fixed confidence setting). Thank you for your suggestion! We have added some intuition in the paper following those comments.
> The idea is that the main source of failure probability for the fixed budget setting is through a mechanism that does not really depend on $\eta$. To succeed in pure exploration, one should have sampled the eventually outputted arm at least $\Omega(\log 1/\delta)$ times. The main obstacle to success in the fixed budget case is that any arm we obtain many samples of might gradually degrade over time. The probability of this degradation is essentially given by small probabilities coming from Chernoff-bound type events, which dominate the prior probability that the arm is in a top quantile.
>
> 2. Thank you for the question! We have added the parameter input line to Algorithm 3, and added clarification in the text at the start of Section 3.2. In particular we referenced Appendix C.1 where algorithms to compute $\alpha$ are given in the three scenarios of Remark 3.1. These were left to the Appendix due to space constraints.
>
> Clarity Questions:
> 1. Thank you for the question! These two approaches of describing sample complexity bounds are just two ways to state the same thing. In our approach, we fix all the other parameters, so for each $N$ there is an optimal $\delta=\delta_N$ that is achievable. To describe the rate at which $\delta_N$ tends to 0, it is equivalent to go the other direction: sending $\delta$ to 0 and asking how large $N=N(\delta)$ needs to be so the failure probability is at most $\delta$. We have added some more explanation to this effect in Section 1.1 Problem Formulation:
> Indeed once $\eta$ and $\varepsilon$ are fixed, the question of minimizing the sample complexity $N=N(\delta)$ (given a target confidence $\delta$) is equivalent to minimizing the failure probability $\delta=\delta(N)$ (given a sample complexity $N$). These viewpoints are equivalent in both settings we study, and we switch between them at times.
>
> 2. Thank you for your suggestion! We have added some clarifications to this remark, which we hope explains things more thoroughly. You are right that the intent was that O(1) batches are sampled in expectation.
> Our fixed confidence algorithm, given by combining Alg. 1 with Alg. 2 as above, requires only O(1) batches in expectation. Here a batched algorithm operates in a small number of batched phases. At the start of each phase, such an algorithm chooses b arms to sample exactly s times each, where b,s can both depend adaptively on the previous feedback, but cannot be changed during the current phase.
>
>
> 3. Thank you for pointing these out! We have fixed these typos from a previous version. Basically, $\eta_1$ and $\eta_2$ were both within a factor 2 of $\eta$, so they are essentially all about the same.
>
> 4. Thank you! We have corrected the typo.

---

> > ### Comment · Reviewer_xhAx · 2023-08-14
> > **Thank you for the rebuttal**
> >
> > Thank you for taking time to answer my queries. I now have a better understanding of the fixed budget setting in the paper. All the best :)

---

### Author Rebuttal · Authors · 2023-08-09

We thank the reviewers for insightful comments and suggestions. We have adopted most of your suggestions and added clarifications in the paper.

Please see the one-page pdf where we highlight some of the more significant modifications so far.

---

### Decision · Program_Chairs · 2023-09-21

**Decision:**

Accept (poster)

**Comment:**

The paper studies the variant of pure exploration for multi-armed bandit --- quantile pure exploration with infinitely many arms. It shows tight results on the fixed-confidence and the fixed-budget settings. All reviewers support the paper and provide constructive comments. The authors' rebuttal addressed many of the comments. I recommend the paper to be accepted by NeurIPS'2023, and wish that the authors could incorporate all the comments from the reviewers into the final version of the paper.